# How EEG preprocessing shapes decoding performance
Roman Kessler [1] ✉, Alexander Enge[1,2] & Michael A. Skeide[1]

Electroencephalography (EEG) preprocessing varies widely between studies, but its impact on classification performance remains poorly understood. To address this gap, we analyzed seven experiments with 40 participants drawn from the public *ERP CORE* dataset. We systematically varied key preprocessing steps, such as filtering, referencing, baseline interval, detrending, and multiple artifact correction steps, all of which were implemented in MNE-Python. Then we performed trial-wise binary classification (i.e., decoding) using neural networks (*EEGNet*), or time-resolved logistic regressions. Our findings demonstrate that preprocessing choices influenced decoding performance considerably. All artifact correction steps reduced decoding performance across experiments and models, while higher high-pass filter cutoffs consistently increased decoding performance. For *EEGNet*, baseline correction further increased decoding performance, and for time-resolved classifiers, linear detrending, and lower low-pass filter cutoffs increased decoding performance. The influence of other preprocessing choices was specific for each experiment or event-related potential component. The current results underline the importance of carefully selecting preprocessing steps for EEG-based decoding. While uncorrected artifacts may increase decoding performance, this comes at the expense of interpretability and model validity, as the model may exploit structured noise rather than the neural signal.

The application of classification models to neural data – known as decoding – has become a standard technique in neurosciences, including electro-encephalography (EEG) research. Unlike univariate approaches, decoding takes advantage of the multidimensionality of the data, facilitating the exploration of basic and translational research questions or the deployment of brain-computer interfaces (BCI). In basic research, decoding can serve as a tool to increase sensitivity[1], to reduce the multidimensional space spanned by electrode voltages to infer the neural representational space, and to quantify how cognitive states, perceptual features, or task conditions are reflected in patterns of brain activity over time and space[2,3]. Translational research uses decoding to better understand individual risk or disease patterns[4] or to mediate relationships between therapy and clinical outcome[5]. On the other hand, BCI studies aim to maximize decoding performance, allowing a user or patient to control a machine or to communicate with the environment[6,7].

Although maximizing decoding performance may not always be the primary goal of a study, it is to some degrees important in contexts known to compromise data quality, such as developmental or patient research where motion is common. However, steps to improve decodability must not sacrifice the interpretability and practical utility of the model. Maximizing decodability can be realized at several levels, such as the experimental design, data acquisition, data preprocessing, or the choice of the decoding framework. In the present study, our aim was to illustrate how certain preprocessing choices applied to data derived from common EEG experimental paradigms can increase or decrease decoding performance.

This research question can be addressed in several ways. For example, a many-teams approach[8–11] can examine the analysis strategies of multiple researchers tackling a set of fixed research questions. Each team thus contributes one analysis pipeline. On the other hand, using a multiverse approach[12,13] one single team contributes many possible pipelines. Unlike a many-teams approach that samples an educated subset of individual pre-processing pipelines (i.e., forking paths) from the community, a multiverse approach allows for a grid search over all possible forking paths by systematically varying each predefined preprocessing step. The multiverse approach is not limited to preprocessing pipelines. It can be applied to any analysis step in a study, such as statistical modeling or the selection of a decoder architecture[14,15].

Similar to previous studies[12,13], we constructed a multiverse for EEG preprocessing (Fig. 1). We systematically varied ocular and muscle artifact correction using independent component analysis (ICA), low-pass filter

---

[1]Max Planck Institute for Human Cognitive and Brain Sciences, Leipzig, Germany. [2]Humboldt-Universität zu Berlin, Berlin, Germany.
✉e-mail: rkesslerx@gmail.com

**Fig. 1 | The multiverse of preprocessing choices.** By systematically varying each preprocessing step, the raw data was replicated 2592 times to be pre-processed in unique forking paths. The asterisks indicate processing choices of an *example forking path* used for some analyses, in which the N170 experiment was re-referenced to the average, and all other experiments were re-referenced to P9/P10.

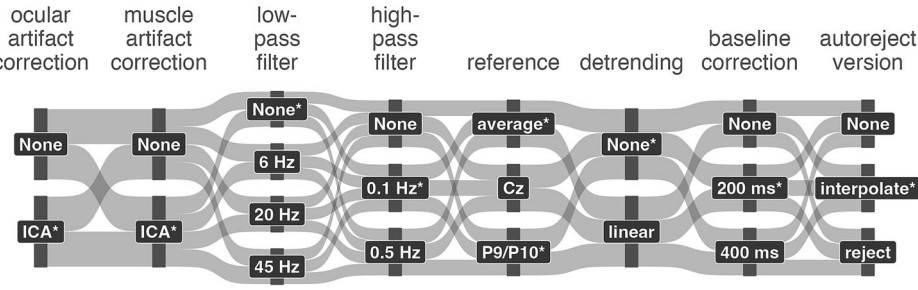

(LPF), high-pass filter (HPF), reference, detrending, baseline interval, and the usage of autoreject[16,17] for artifact rejection. In contrast to previous multiverse studies, we did not examine the impact of preprocessing on event-related potential (ERP) amplitude but on decoding performance. To this end, we analyzed several openly available EEG experiments[18,19] with classifiers derived from two frameworks. First, we used a neural network-based classifier (EEGNet)[20], that extracts temporal and spatial features for each entire trial to predict the experimental condition, i.e., stimulus or response. Second, we used time-resolved logistic regression classifiers[21,22], that use the electrode signals at each time point in isolation to predict the experimental condition.

Our results demonstrate how preprocessing steps influence decoding performance. Narrow filters increased the decoding performance, especially in a time-resolved framework. In contrast, artifact correction steps generally decreased the decoding performance. We discuss promises and pitfalls in EEG preprocessing with respect to decoding analyses.

## Results

### Evoked responses
Grand average evoked responses were computed for each experiment of the *ERP CORE* dataset[18,19] (Fig. 2, Table 1) using one example forking path, i.e., one particular preprocessing pipeline (Fig. 1). The time courses of the components closely followed the results of the original study (see ref. 18 and their Fig. 2), despite differences in preprocessing, the inclusion of all participants, and the lack of manual intervention throughout the preprocessing. The components differed in amplitude between experiments. The error-related negativity (ERN) and N400 showed the largest amplitudes, whereas lateralized readiness potential (LRP), mismatch negativity (MMN), and N2pc showed rather small amplitudes. Furthermore, the shapes of the components differed, with N400 and P3 covering large time intervals, while, N2pc, for example, covered only a short time interval.

### Decoding performance
Decoding was performed separately for each forking path using a neural network-based approach (EEGNet) and a time-resolved approach using logistic regression. For EEGNet, we quantified the (balanced) test accuracies of each forking path – averaged across participants – as a measure of decoding performance (Fig. 3A). For time-resolved decoding, we quantified $T$-sums across participants but within a forking path as a measure of decoding performance (Fig. 3B). Different decoding performances across experiments were observed, with the conditions in ERN being easiest to decode with a median accuracy of roughly 0.85, followed by LRP, followed by all others, and lastly MMN with a median accuracy of roughly 0.57. The ranking of the decoding performances was similar for the neural network-based and the time-resolved approaches. When the time-resolved decoding performance was quantified using the average decoding accuracy over a time window instead of the $T$-sum (Fig. S1), the ranking of the experiments remained similar.

High ERP amplitudes at single electrodes were not necessarily associated with high decoding performance across the scalp. For example, LRP showed a rather low amplitude at the selected electrodes (Fig. 2), while the decoding performance was rather high (Fig. 3). Although the two measures

are not directly comparable, high amplitudes and long duration of ERPs at one electrode could still facilitate decoding performance across all electrodes, as the data underlying the ERP are also incorporated into the classifier, especially if the high amplitudes are the result of low variability between trials. Furthermore, the ranking not only depends on the magnitude and extent of a component, but also on characteristics of the experimental design, such as the number of stimulus repetitions. Note that the $T$-sum values depend on the data sampling rate, which was held constant between experiments. Therefore, the exact values are rather arbitrary.

We marked the forking path without any preprocessing, i.e., no artifact correction, no filtering, no baseline correction or detrending, and a common online reference Cz (Fig. 3), while keeping the implemented data normalization of the respective decoder (c.f. Methods). For EEGNet, the forking path without preprocessing performed well above average, indicating high flexibility regarding the shape of the features. For time-resolved decoding, unprocessed data performed badly, indicating that at least minimal preprocessing is required for decoding.

Figure 4 illustrates the time-resolved decoding performance of all seven experiments for one single forking path (Fig. 1). Decoding time series revealed large, significant clusters, often spanning time windows of over 600 ms. Peak decoding accuracies were roughly between 0.55 (MMN) and 0.7 (ERN). The significant decoding clusters tended to span larger time intervals than the ERPs on selected channels (Fig. 2). Note that exact cluster on- and offsets are not interpretable using the performed cluster mass test[23], as compared to a cluster depth test[24].

### Influence of preprocessing choices
For each experiment, we constructed separate linear mixed models (LMMs) using EEGNet decoding accuracies or linear models (LMs) for $T$-sums derived from time-resolved decoding, describing the decoding performance as a function of all preprocessing steps. One model was fitted per decoding framework and experiment, resulting in seven separate LMMs (EEGNet) and seven separate LMs (time-resolved). Within each model, we then estimated the marginal means for each preprocessing step, i.e., the influence of each choice for a given step on the decoding performance, marginalizing out all other steps. Figure 5 illustrates the marginal means of the main effects for all experiments, both types of decoding models, and for all possible variations of each preprocessing step. The influence – in percent deviation from the marginal mean – of the preprocessing choice on the decoding performance was larger in the time-resolved analysis than in the EEGNet analysis. This is partly due to the deployed $T$-sum metric and the fact that the time-resolved analysis is performed across participants. See Fig. S2 for results using average accuracies over time and LMMs for time-resolved decoding.

The impact of most processing steps followed the same direction across experiments (Fig. 5). For example, using artifact correction steps such as ICA or the autoreject package decreased decoding performance in most experiments and decoding frameworks. Furthermore, a high HPF yielded the best decoding performances across experiments and decoding frameworks. In the time-resolved framework, a low LPF further increased decoding performance, while no such trend was observed in the EEGNet framework. The choice of a reference maximizing performance varied between experiments and model types, but its influence was rather weak

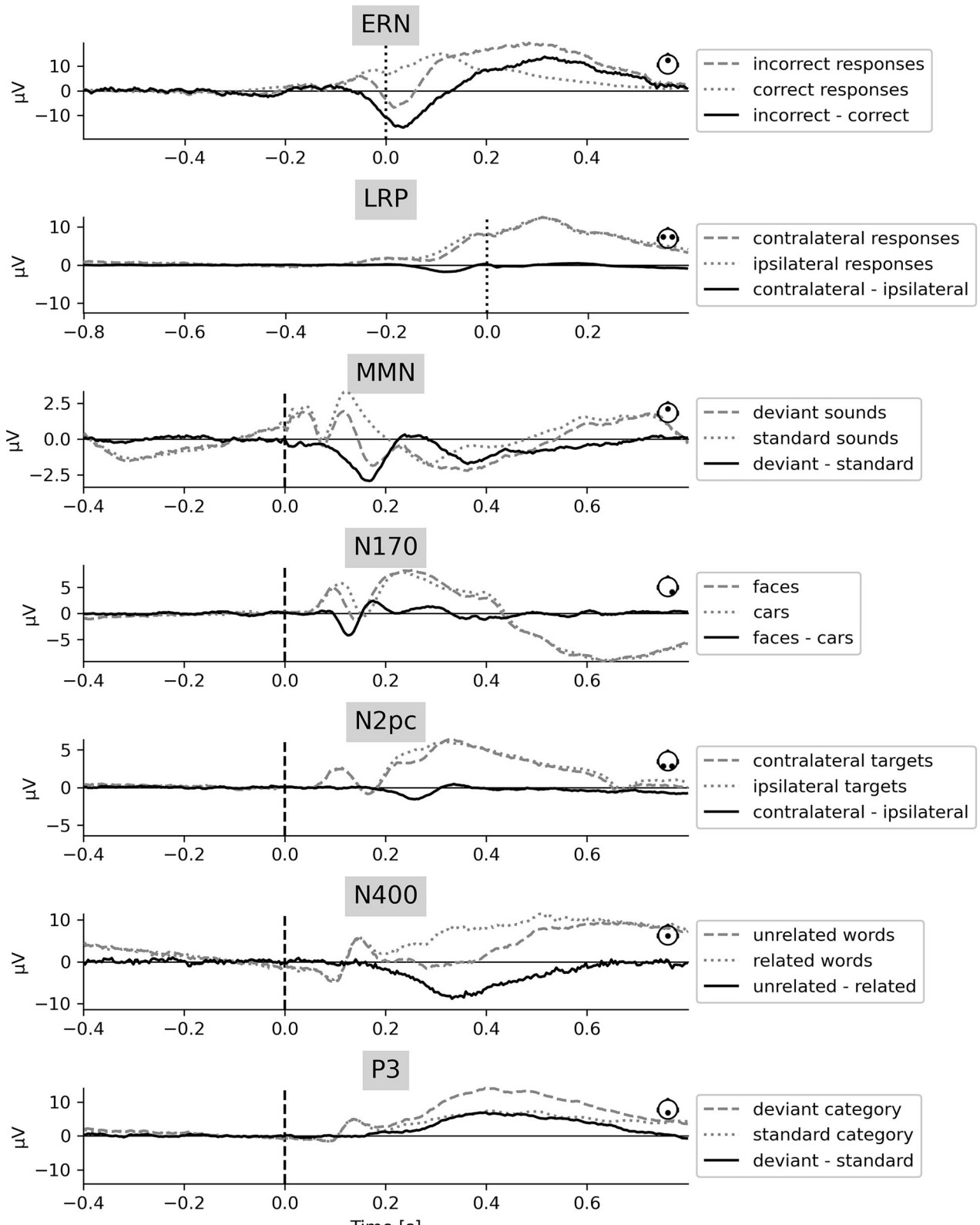

**Fig. 2 | Grand average evoked responses for each experiment.** The data from one example forking path (Fig. 1) were used. The dashed and dotted time courses represent the average responses to each stimulus category. The solid time course represents the difference between the respective categories, illustrating the respective ERP. Time series originate either from a single channel, or from two channels, in which case the mean was calculated. The channel positions are indicated by small graphical legends on the right of each plot. Dotted vertical lines indicate response onset (ERN, LRP), and dashed vertical lines indicate stimulus onset (other experiments). Note that the y-axes are scaled differently for each experiment.

## Table 1 | Properties of the ERP CORE dataset

| Experiment/ component | Paradigm | Conditions | | Channels evoked |
|---|---|---|---|---|
| | | **Evoked** | **Decoding** | |
| ERN | Eriksen flanker task[40] | Incorrect & correct responses | Incorrect & correct responses | FCz |
| LRP | Eriksen flanker task[40] | Contra- & ipsilateral responses | Left & right responses | C3/C4 |
| MMN | Passive auditory oddball | Deviant & frequent sounds | Deviant & frequent sounds | FCz |
| N170 | Face perception | Faces & cars | Faces & cars | PO8 |
| N2pc | Visual search | Contra- & ipsilateral targets | Left & right targets | PO7/PO8 |
| N400 | Word pair judgment | Unrelated & related words | Unrelated & related words | CPz |
| P3 | Active visual oddball | Deviant & frequent category | Deviant & frequent category | Pz |

The experiments are named after the main component, analyzed with the respective paradigm[18]. For each experiment, the conditions and channels used to compute and visualize the ERPs are shown, with the second condition being subtracted from the first condition. For decoding, the trials of experiments LRP and N2pc were assigned to different conditions.

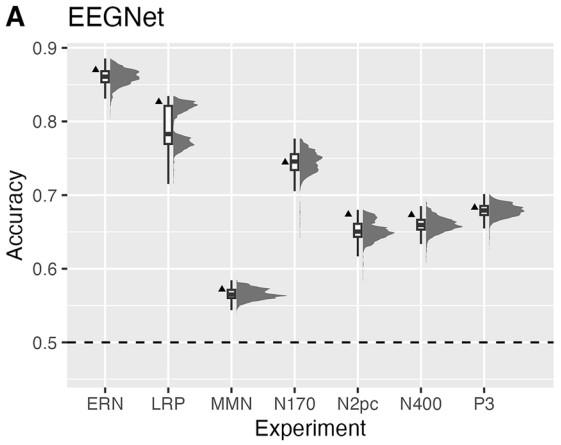
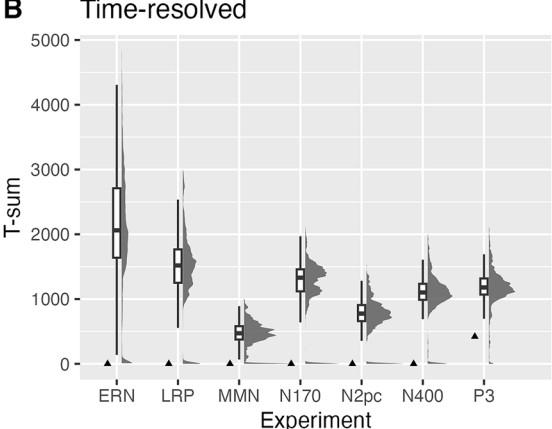

**Fig. 3 | Overview of decoding performances. A** EEGNet: (Balanced) Decoding accuracies (y-axis) are plotted for each forking path, averaged across participants, separately for each experiment (x-axis). **B** Time-resolved: *T*-sums (y-axis) are plotted for each forking path and across participants, separately for each experiment (x-axis). Triangles indicate the forking path without preprocessing. Boxes represent the interquartile range (25th to 75th percentile), with the median indicated by a solid black line. Whiskers extend to the most extreme values within 1.5 times the interquartile range from the lower and upper quartiles.

compared to other preprocessing steps. Linear detrending had a rather positive effect on decoding performance for most experiments and decoding frameworks. Furthermore, a longer time window for baseline correction was beneficial for decoding performance in most experiments. Tables S1 & S2 illustrate the results of omnibus *F*-tests for all preprocessing steps, experiments, and model types. In addition, tables S3 & S4 illustrate the significant pairwise post-hoc tests between factor levels of the main effects.

Two peculiarities emerged, particularly pronounced for EEGNet (Fig. 5A). First, the removal of ocular artifacts was strongly negatively associated with decoding performance in the N2pc experiment. In this experiment, eye movements are expected since the participants might perform involuntary saccades into the visual hemifield of the target, and the target position was decoded (Table 1). Therefore, ocular artifacts are expected to be systematically associated with the class label and thus predictive for the decoder. Accordingly, removing these artifacts reduced decoding performance. Second, removing muscle artifacts was negatively associated with decoding performance, especially in the LRP experiment. In this experiment, left and right hand button presses were decoded (Table 1), and thus, systematically different muscle artifacts are expected for the two conditions. These artifacts were predictive, and therefore removing them decreased decoding performance.

Further, we analyzed the impact of changing one single preprocessing option at a time compared to the example forking path (Fig. 1) on decoding performance. Figure S3 illustrates that for most experiments, leaving out artifact correction steps (ICAs, autoreject) increased decoding performance. Similarly, the 6 Hz LPFs increased decoding performance in time-resolved decoding. For the other preprocessing steps, a change compared to the

reference forking path has different consequences for each experiment. The comparison gives insights only in how a punctual substitution of one (but not more) preprocessing step in the reference forking path impacts decoding performance, and is therefore highly dependent on the reference forking path, which in turn varies largely across researchers[25]. To allow for a simple comparison of individual preprocessing pipelines and their impact on decoding performance, we deployed an online dashboard at https://multiverse.streamlit.app.

We ranked all forking paths based on their average decoding performance across participants from best (#1) to worst (#2592), separately for each decoding framework and experiment. Figure S4 & S5 illustrate the individual preprocessing choices of the performance-ranked forking paths and provide more detail on the data points of Fig. 3. It turns out that in many experiments analyzed with EEGNet, the forking paths with the highest decoding performance did not include muscle artifact correction. Especially in LRP, this effect explains the bimodal distributions seen in Fig. 3 by suggesting that all forking paths using muscle artifact correction were ranked lower. Furthermore, for EEGNet, the forking paths, including autoreject, showed lower decoding performance. For N2pc, the forking paths with the highest decoding performance did not include ocular artifact correction. For time-resolved decoding, a positive effect of detrending and baseline correction on decoding performance was observed in most experiments.

### Interactions between preprocessing choices

We also analyzed two-way interactions between the preprocessing steps. Figure 6 illustrates the interactions related to the N170 experiment within

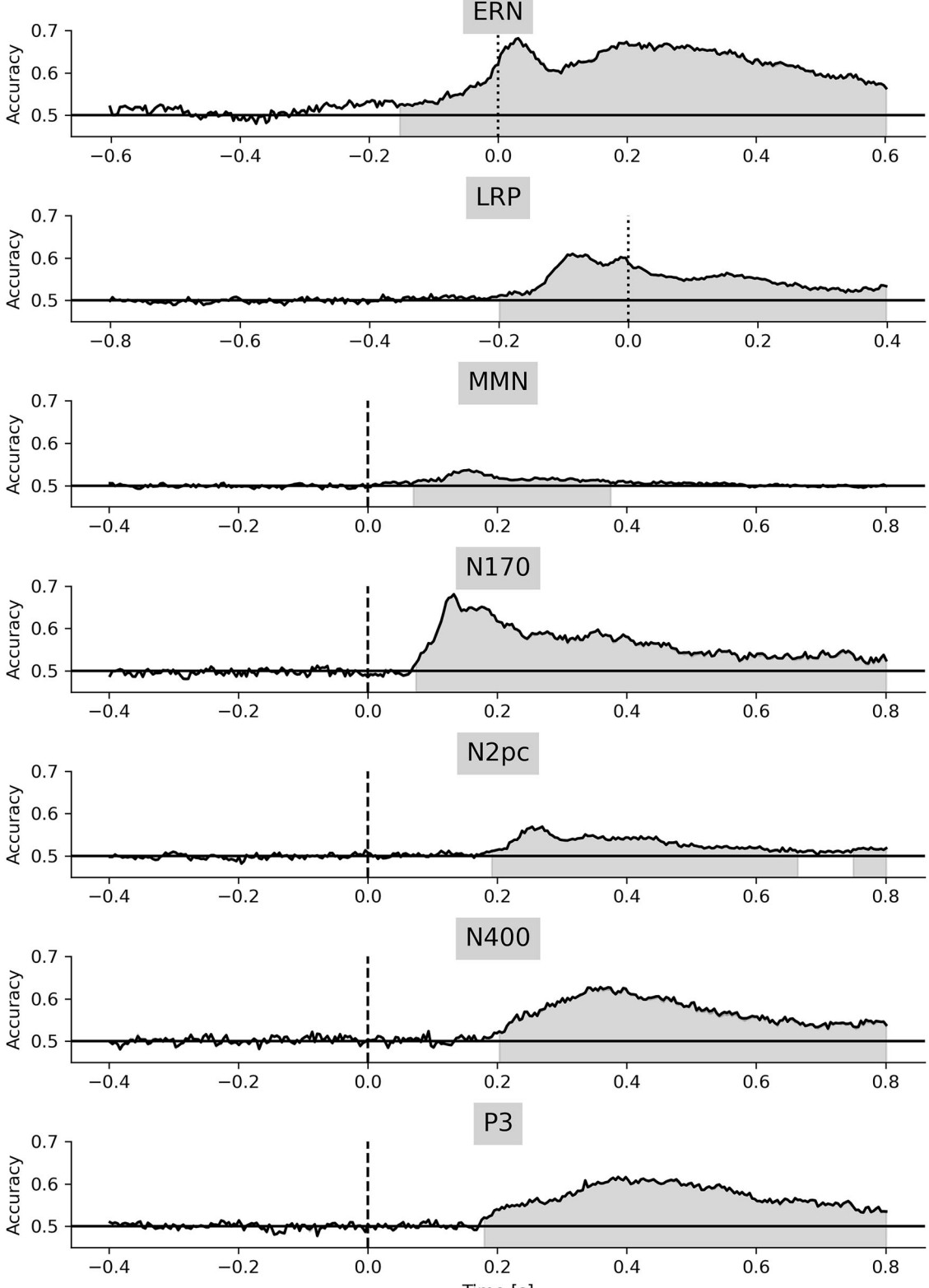

**Fig. 4 | Time-resolved decoding accuracy for one forking path.** (Balanced) Decoding accuracies are illustrated on the y-axis for each time point (x-axis). The horizontal black line represents the chance level. One example forking path was used per experiment (Fig. 1). Decoding was performed within each participant, but the decoding time series were averaged across participants. Different experiments are separated in vertical panels. Dotted vertical lines indicate response onset (ERN, LRP), and dashed vertical lines indicate stimulus onset (remaining experiments). Permutation cluster mass tests with a family-wise error rate correction at $\alpha = 0.05$ were performed for each experiment (shaded areas).

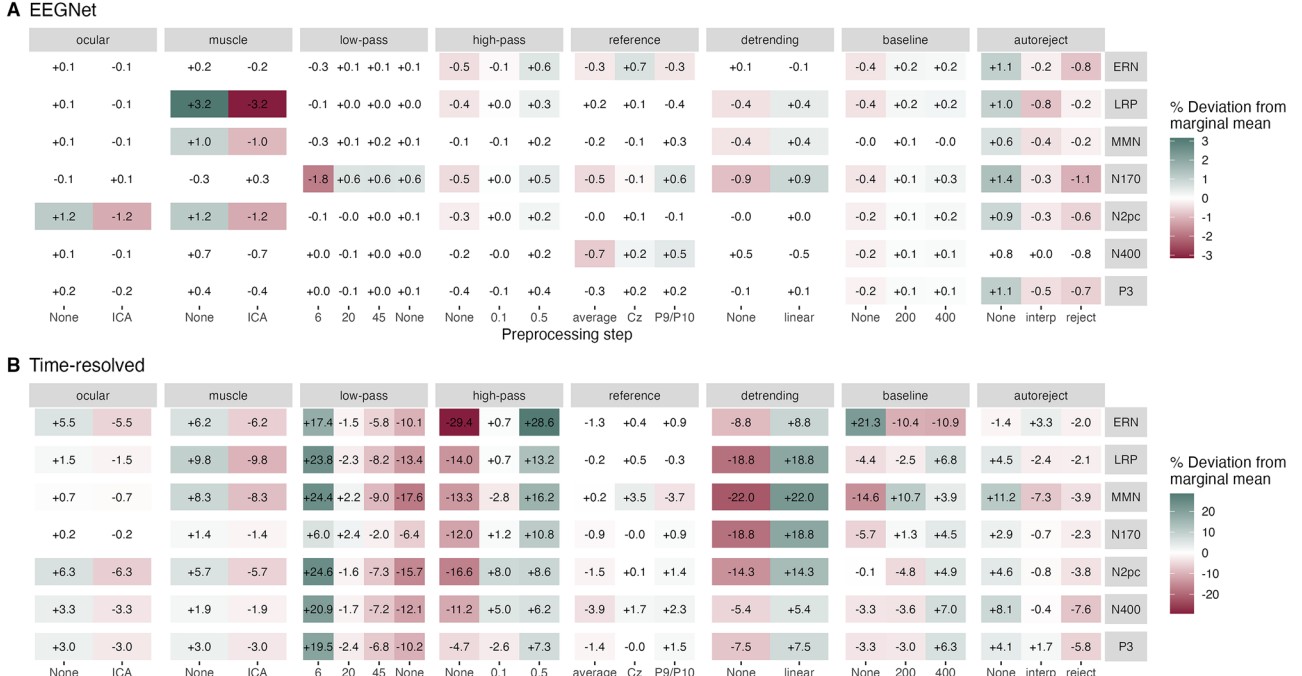

**Fig. 5 | Influence of preprocessing steps on decoding performance.** Percentage deviation from marginal means of either decoding accuracy (EEGNet, **A**) or *T*-sum (time-resolved, **B**) are depicted within each tile. Marginal means for each level (x-axis) of preprocessing step (horizontal panels) are normalized to the mean of the respective experiment (vertical panels). Each tile therefore shows the percentage differences in relation to this mean value. Only steps with a significant *F*-test ($p < 0.05$) are colored. Color scales differ in **A** and **B**. *Ocular*: ocular artifact correction; *muscle*: muscle artifact correction; *ICA*: independent component analysis; *low-pass*: low-pass filter in Hertz; *high-pass*: high-pass filter in Hertz; *baseline*: baseline interval in milliseconds; *autoreject* version either interpolate (*interp*) or reject artifact-contaminated trials (*reject*).

the EEGNet framework as an example. Due to a considerable number of combinations of experiments and decoding frameworks, we illustrate the remaining interaction results in the Supplements (Fig. S6–S18).

Strong interactions can be observed between HPF and detrending (Fig. 6). In many experiments, applying linear detrending made the choice of the HPF less important, i.e., neutralized an adverse choice regarding the HPF cutoff (Tables S1 & S2, Fig. S6–S18). Similarly, for N170 and other experiments, a combination of neither applying HPF nor baseline correction had negative effects on decoding performance.

Several effects can be observed when using autoreject in the *reject* version, in which noise-contaminated trials were discarded rather than interpolated (c.f. Methods). Decoding performance was lower when autoreject was used, but either HPF, linear detrending, or baseline correction was omitted (Fig. 6, S4 & S5). Presumably, the high electrode voltages – likely present when none of these steps were applied – led to the rejection of many trials, and thus fewer trials were available for model training.

Two pronounced interactions were detected for time-resolved decoding. First, there was an interaction between detrending and baseline correction, with detrending being critical when baseline correction was avoided (Fig. S12–S18). Second, although higher-order interactions were not included in the models, forking paths which included neither baseline correction, detrending, nor HPF showed the lowest decoding performance (Fig. S4 & S5). Tables S1 & S2 and the corresponding Fig. 6 & S6–S18 illustrate all interaction effects for the other experiments and decoding frameworks. Some interactions were common across experiments, while others could only be observed in single experiments.

**Significant clusters in time-resolved decoding**
Decoding time series for each forking path and experiment were also visualized individually and ordered according to the total *T*-sum. Figure S19 illustrates the decoding accuracy for each time point that fell within a

significant cluster. The peak decoding accuracies were mainly in a similar time window and at a similar amplitude, regardless of the forking path. In most experiments, the extent of the significant clusters remained largely similar for the roughly 90% highest-ranked forking paths (Fig. S19). The significant time windows narrowed down or vanished completely for the 10% lowest-ranked forking paths. Most of the forking paths revealed time windows in which conditions were decoded significantly.

Some decoding time series, especially those corresponding to the higher-ranking forking paths, showed significant clusters extending into the baseline time window, resulting in a baseline artifact. While a certain percentage of false positives was expected across all forking paths in each experiment, some experiments exhibited a higher amount of spurious decodability, extending into baseline periods. Note that one-third of forking paths did not use baseline correction (Fig. 1).

We compared forking paths with and without baseline artifact, and scrutinized the distinctive preprocessing steps of these forking paths. Therefore, we defined baseline artifacts as significant clusters extending into the baseline period (Table S5). Figure S20 illustrates how many forking paths, including a particular version of a preprocessing step, led to a baseline artifact in the respective experiment. It turns out that experiments with large ERP components (see Fig. 2), such as ERN, N400, and P3, were more likely to show baseline artifacts. For most experiments, linear detrending increased the likelihood of a baseline artifact (Fig. S20). A low LPF (6 Hz), omitting baseline correction, and in some cases a high HPF (0.5 Hz), increased the likelihood of a baseline artifact (Fig. S20). Artifact rejection steps tended to decrease the probability of a baseline artifact in most experiments (Fig. S20), highlighting the importance of artifact rejection in rendering the decoding results based on neural signals rather than physiological artifacts. One likely contributor to baseline artifacts is the non-causal filters used in the present study, which allow the signal to affect earlier time points[26].

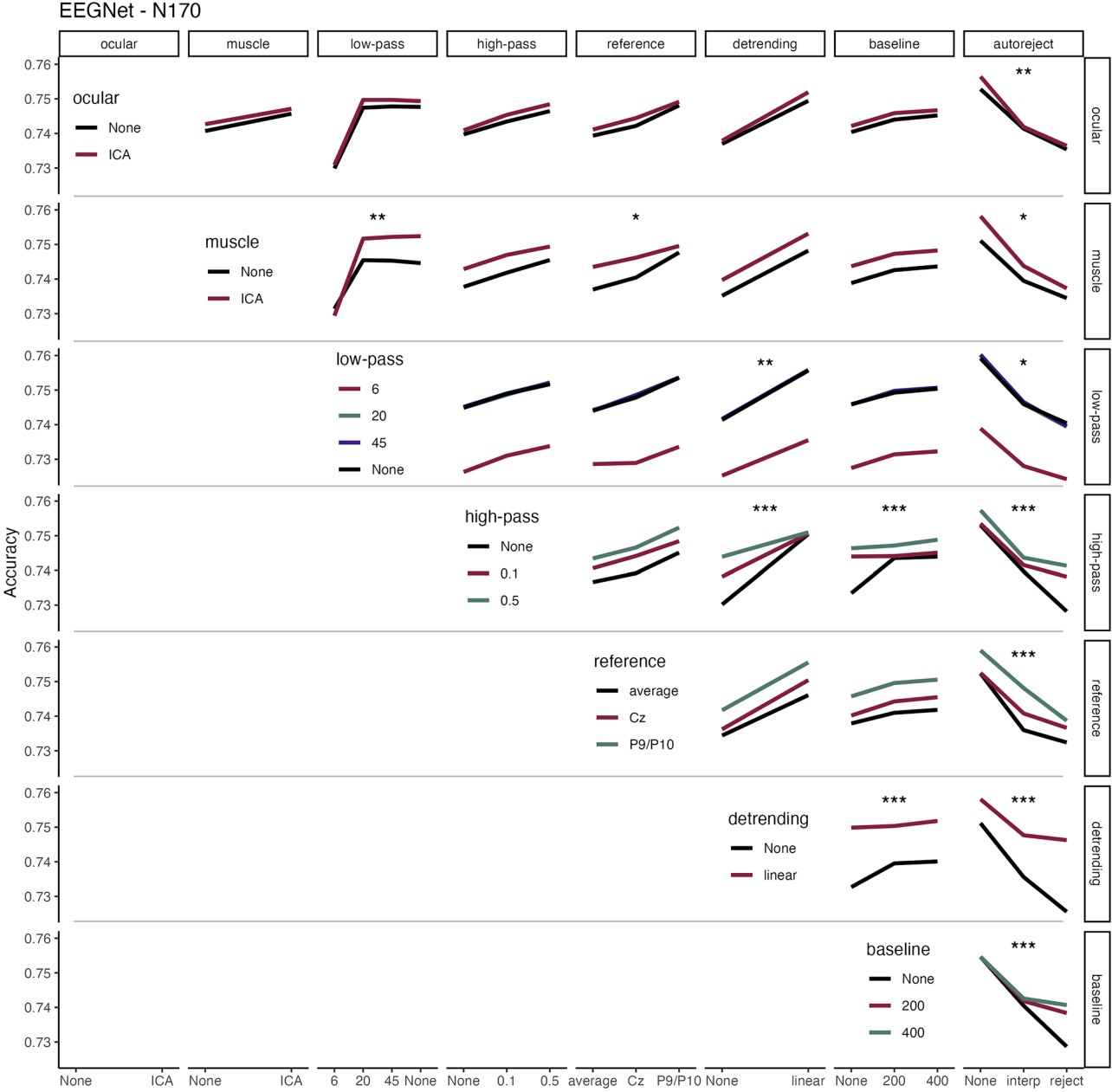

**Fig. 6 | Interactions between preprocessing steps on decoding performance for the N170 experiment and EEGNet decoding.** Horizontal and vertical panels illustrate the different preprocessing steps. The individual preprocessing choices are illustrated on the x-axes and color-coded. The color legends on the diagonal refer to each horizontal panel. (Balanced) Decoding accuracies are shown on the y-axes. Stars indicate the significance ('*' $p < 0.05$; '**' $p < 0.01$; '***' $p < 0.001$). *Ocular*: ocular artifact correction; *muscle*: muscle artifact correction; *ICA*: independent component analysis; *low-pass*: low-pass filter in Hertz; *high-pass*: high-pass filter in Hertz; *baseline*: baseline interval in milliseconds; *autoreject* version either interpolate (*interp*) or reject artifact-contaminated trials (*reject*).

## Influence of the participant

To demonstrate the variability of EEGNet decoding performance across forking paths (Fig. 3A), we averaged across participants to be consistent with Fig. 3B, since the *T*-sums presented there were calculated across participants. Without averaging, the variability in accuracy was considerably increased, highlighting a large effect of the participant on decoding accuracy compared to the effect of the forking path (Fig. S21). The remaining variation in decoding performance in each experiment in Fig. 3 can be attributed to the choice of the forking path.

We further analyzed whether decoding performance was associated with participant demographics, but did not find an association between age, sex, or handedness and participant-specific decoding performance according to the LMMs' random intercepts (Fig. S22 & S23, Supplementary Text – Influence of the participant in the LMMs). In addition, we correlated the participant-specific random intercepts across experiments, but we only found a significant linear correlation ($r = 0.52$, $p = 0.011$, false discovery rate-corrected) between LRP and N170 in the EEGNet framework (Fig. S24 & S25). While the participant's influence on decoding performance is substantial within individual experiments, there is no consistent evidence that the same participant achieves higher decoding performance across multiple paradigms. This aligns nicely with a previous study, in which a different neural network decoder and a single forking path were used on the very same data set[27].

## Discussion

In this study, we investigated the influence of EEG preprocessing steps on decoding performance in two different decoding frameworks. For the neural

network-based framework (EEGNet), we compared test accuracies of models fitted to differently preprocessed epochs. For the time-resolved framework, we computed cluster-based permutation statistics on the decoding time series across participants for differently preprocessed epochs and quantified decoding performance as the *T*-sum of the significant clusters. We analyzed seven different experiments separately and modeled the classifier performance as a function of all preprocessing steps.

The results demonstrated that above chance-level decoding was feasible with most forking paths. For EEGNet, high HPF cutoffs and baseline correction facilitated decoding. For time-resolved classifiers, linear detrending, low LPF cutoffs, and high HPF cutoffs increased decoding performance. All artifact correction steps reduced decoding performance for all experiments and decoding frameworks, likely by removing artifacts that covaried with the experimental condition, but potentially also by removing portions of the neural signal of interest[28]. Hence, focusing on decoding performance alone is not sufficient to decide on a preprocessing pipeline, but requires further investigation in the light of neural signal versus noise. We also found detrimental combinations of processing steps that decreased decoding performance substantially.

### Artifacts spuriously enhance decoding performance

Most artifact corrections led to a decrease in decoding performance. This decrease was particularly pronounced in experiments where artifacts can be expected to covary with stimulus categories, such as ocular artifacts in a visual search task and muscle artifacts in a keystroke task. Despite the decrease in performance, removing artifacts should be of interest when model features are to be interpreted spatially or temporally, such as could be done by means of Shapley values[29] or by visualizing weight vectors[30]. However, removing artifacts may also remove the neural signal of interest[28], especially when using the threshold-based ICA employed in the present study. Future studies should further disentangle the influence of artifact correction steps into the unique contribution of the removed artifacts versus the unintentionally removed neural signal. In some BCI use cases in which the source of the signal may be less relevant, an analyst may even refrain from removing predictive artifacts.

Compared to classical ERP analysis, decoding uses all channels and time points simultaneously to infer differences between conditions[1]. This can lead to a situation in which certain artifacts, while less problematic for ERP analyses in a region of interest, may drive whole-brain decoding and thus inadvertently imply stimulus-related differences. BCI studies have also suggested a drop in decoding performance when various artifact correction methods were applied[31,32].

### Narrow filtering yields high performance

Filter cutoffs also play an important role in decoding performance. Models using data filtered with high HPF and low LPF cutoffs ranked highest in all experiments using time-resolved decoding. This finding supports using a low LPF cutoff to exclude alpha activity, an influential source of trial-to-trial variability[33]. However, a low LPF cutoff may inflate the time window of an ERP and thus preclude the interpretation of its timing[34]. The same should hold for the temporal interpretation of decoding accuracies. On the other hand, a LPF could partially mitigate a missing muscle artifact rejection step, since muscle artifacts are largely distributed in a frequency spectrum above 15 Hz[35].

A high HPF cutoff increased decoding performance in both decoding frameworks. HPFs are important for leveling the signal of each epoch, especially when baseline correction has not been performed. However, HPF cutoffs above approximately 0.3 Hz also come at a cost, introducing artifacts of opposite polarity before and after the ERP[36], which in turn compromise the temporal interpretation of decoding performance. Designing more specified filters can decrease such artifacts[26], including some of the baseline artifacts seen in the current study. In particular, the use of non-causal filters allows the signal to leak into preceding time points, increasing the likelihood of a baseline artifact. Linear detrending was also positively associated with decoding performance, especially for time-resolved decoding. Before

classifier training, the voltage values at each time point and channel were standardized across trials for time-resolved decoding. Random drifts in some electrodes and trials may generate relatively large values – especially at epoch margins – obscuring potentially predictive values of other epochs situated at a much smaller scale at the same electrodes and time points. Only a few other decoding studies also removed these remaining drifts from the epochs (e.g., ref.[37]).

### Detrimental combinations of preprocessing steps

Unlike previous studies[12,13], we also allowed for interactions between preprocessing choices. The most common interactions across experiments and decoding frameworks were found between preprocessing steps that level the signal of an epoch to a common range. These include HPF, detrending, and baseline correction. Combining different versions of these steps that do not level the signal to a similar range, i.e., no HPF, no detrending, and no baseline correction, tended to reduce decoding accuracy. Not leveling the signal by means of these preprocessing steps also led to high trial dropout rates when using autoreject.

Furthermore, spurious decodability in baseline periods was observed in many forking paths, leveraged by detrending, a low LPF, a high HPF, or the avoidance of baseline correction. Another study also demonstrated spurious baseline decodability by the combination of a high HPF, baseline correction, and avoidance of robust detrending[38]. Such an effect may also be present in the EEGNet approach, as the baseline was also included in the training of the classifier. We would therefore suggest to also decode from the baseline in the time-resolved framework – and visualize baseline decodability – for the sake of detecting such artifacts. If the baseline period is not decoded in the time-resolved framework, spurious baseline decodability and the likely associated modification in the to-be-interpreted window of the decoding time series might evade detection. Artifact correction steps however decreased the likelihood of baseline artifacts.

### 'Optimal' preprocessing differs for ERPs and decoding

Previous studies using multiverse-like preprocessing have mainly focused on 'optimal' preprocessing with respect to ERP characteristics such as amplitude. Since one cannot directly distinguish between neural signal and artifacts covarying with the experimental condition, focusing on maximizing amplitude (or similar) has similar limitations as focusing on decoding performance (accuracy or *T*-sum) alone. Results from multiverse studies need to be enriched with considerations about the origin of the signal, the side effects of individual processing steps in general, and domain knowledge about the processes under investigation. However, multiverse analyses on single outcome metrics are a helpful step towards a better understanding of the impact of preprocessing and, among other possibilities, facilitate the comparison of different study results. The following is a comparison of how different the effects of preprocessing can be on ERP-related measures versus decoding performance.

For example, a study by Delorme[39] compared preprocessing pipelines and their influence on the number of significant channels between two conditions in three paradigms (Go/No-go task, face perception task, & active auditory oddball task with button press). Their optimal pipeline was dependent on the software used for preprocessing, but mainly comprised a HPF and bad channel interpolation. Other steps either reduced or increased the number of significant channels or did not show any effect. While their study tested a wider range of artifact correction techniques, the different directions of effects on the number of significant channels in the analyzed experiments make comparison more challenging. In our experiments artifact correction rather decreased the decoding performance. Both studies however, align on the use of HPFs, as in all datasets, higher HPFs increased the number of significant channels or decoding performance.

A study by Clayson et al.[13] employed a multiverse to compare the influence of preprocessing on ERP amplitude for two components, ERN and error-related positivity (Pe), using the Eriksen flanker task[40]. For ERN, their optimal processing consisted of a 15 Hz LPF and ocular artifact correction with ICA. For Pe, however, their optimal processing in terms of ERP

amplitude included a 30 Hz LPF, a 0.1 Hz HPF, and regression-based ocular artifact correction. Our time window for decoding in the ERN experiment spanned the time windows of both components. We however, defined a stimulus-locked rather than a response-locked baseline window[41]. In our search space, the optimal LPF cutoff was at 6 Hz for time-resolved decoding. In contrast, for the neural network-based framework, all higher LPF cutoffs ($\geq$ 20 Hz) were equally suitable for decoding. We found the optimal HPF cutoff to be around 0.5 Hz in both of our decoding frameworks, while Clayson et al. did not test beyond 0.1 Hz. Ocular artifact correction played a minor role in decoding, but avoiding this step still resulted in higher decoding performance in our data. While they found a mastoid reference to be beneficial for ERN amplitude compared to an average reference, we did not find a significant improvement (Table S3). However, we found that the Cz reference increased decoding performance (Fig. 5, Table S3). Because Cz is close to FCz, which itself shows high ERN amplitude (Fig. 2, Table 1)[13,18], subtracting Cz from the remaining electrodes likely injects this signal into the other channels, again limiting potential downstream interpretability of the model weights.

Šoškić et al.[12] analyzed the influence of preprocessing on the effect magnitude in an N400 experiment. The largest effect was obtained with an HPF between 0.01 Hz and 0.1 Hz (compared to 1 Hz) and a mastoid reference (compared to an average reference). We found the same optimal reference for decoding in the N400 experiment for both neural network-based and time-resolved decoding. However, we found that higher HPFs between 0.1 Hz and 0.5 Hz led to higher accuracies.

## Generalizability of the insights from the multiverse

Some of the observed effects of preprocessing on decoding performance were generalizable across experiments, such as the LPF cutoff (time-resolved), baseline correction (EEGNet), or detrending (time-resolved). Some effects were even generalizable across decoding frameworks, such as the HPF cutoffs or the drop in decoding accuracy by artifact correction steps.

The generalizability of the results to different decoding frameworks remains to be demonstrated in the future. We deliberately used two different decoding frameworks to show similarities and differences regarding the influence of preprocessing using multiverse analyses. However, even changing the decoding model – e.g., to support vector machines in a time-resolved framework, or to different neural networks instead of EEGNet – could change the multiverse results. Decoder hyperparameter optimization may also change the optimal preprocessing choices, but this topic is beyond the scope of the current study. Instead, we used default hyperparameters. Decoder hyperparameter optimization could be added to the multiverse, as could the choice of the decoding model itself. One could even treat the multiverse for preprocessing as a kind of grid search-like hyperparameter optimization, if cross-validated and applied to independent test data.

The experiments used in the present study were tailored to ERP-related research questions. ERPs have marked voltage differences in defined channels and time windows, which are likely to also become predictive in decoding. Moreover, the influence of the forking path might be larger if, for example, the experiments included fewer repetitions per stimulus category. Such an effect has been demonstrated previously, illustrating that artifact correction steps have larger effects when a smaller number of trials is analyzed[32]. Good decoding performance was expected for the well-replicated effects studied here.

However, different kinds of EEG paradigms need different decoding frameworks. For example, for experiments in which the spectral power differences between conditions are analyzed[42,43], a decoding framework tailored to frequency decoding is more suited[44]. Recent experiments are often based on the rapid serial visual presentation (RSVP) structure[45–47], in which many different object categories are presented in quick succession to run a large number of trials per object category. These experiments, also employed in the auditory domain[48,49], are not directly designed to find well-described ERPs. Instead, the literature describes ERPs only for a small subset of the contrasts that can be constructed between categories included in an RSVP experiment. For example, faces can usually be easily distinguished

from other categories in decoding based on the data underlying the N170, but also information from other channels not typically used in characterizing the N170. However, differences between other categories (e.g., computers and cars) are less well-described. A possible explanation is that the differentiation of these categories had little evolutionary relevance, leading to idiosyncratic representation for each participant rather than a systematic and generalizable representation across participants. More dense electrode coverage might be helpful to find such subtle differences between categories. From this point of view, the decoding accuracies shown in the present N170 experiment presumably represent the upper limit of decoding accuracies that can be expected from such an RSVP experiment at the given number of stimulus repetitions, due to the strong nature of the N170 component. Other contrasts might lead to classifiers with lower accuracy. Importantly, one should not extrapolate the present recommendations based on the multiverse results regarding the N170 experiment – the only object categorization experiment analyzed in the present study – to an RSVP experiment. The reason is that the preprocessing would then be optimized for the face vs. car contrast, but not for the many other contrasts between categories. In fact, any preprocessing prior to multi-class decoding could bias decodability by filtering out the predictive features of one class represented in a particular frequency spectrum. From this perspective, using the data in a rather raw (i.e., not heavily preprocessed) form might be favorable in this particular context, given that the classifier works well with it.

Finally, the multiverse also illustrates the challenge of analytical variability with regard to preprocessing. Although only a subset of possible forking paths would be practically used by an analyst, the variability is still substantial[25], offering an analyst sufficient degrees of freedom. Preregistration should be one important building block to increase transparency and avoid the potential for malpractice[50,51].

## Inflating the multiverse

The constructed multiverse naturally included only a limited subset of preprocessing steps and variations. One could have added any number of additional variations of each analysis step. For example, one could have sampled the filter cutoffs more densely to find a better optimum, rather than just an optimum based on the available choices. Similarly, the effect on decoding performance of using GLM-based rather than subtraction baseline correction, which has been shown to reduce artifacts in the ERP window, could be investigated[52,53]. Furthermore, a large number of plausible methods have been published for each preprocessing and artifact correction step, such as regression-based methods for ocular artifact correction[54], trial-masked robust detrending[38], robust averaging[55], or the application of pre-trained classifiers on independent components[56], to name a few. For this reason, we have made some simple but reasonable choices for each step that were readily available in the *MNE* package, such as threshold-based ICA for ocular artifact correction. It remains to be seen whether these choices could generalize to similar approaches, other software, and other package versions. For artifact rejection tools that remove the respective artifacts more thoroughly – while not removing neural signal of interest[28] – we expect the reported effects to be even stronger.

Besides the exact steps performed, previous studies also vary widely in their order of preprocessing steps[8]. Systematically varying the order of the steps however, leads to a combinatorial explosion, complicating both computation and interpretation. Furthermore, the interpretation of interactions between preprocessing steps changes depending on their order. Accordingly, such a study needs to be conducted in the future. We have tested one alternative multiverse with a slightly different order of steps, i.e., (1) re-referencing, (2) HPF & LPF, (3) ocular artifact correction, (4) muscle artifact correction, (5) baseline correction & detrending, and (6) autoreject (Fig. S26 & S27, Supplementary Text – Alternative order of preprocessing steps). Most effects were qualitatively similar to those presented in the main manuscript.

A wide variety of preprocessing and artifact correction methods are available from the literature and the EEG community, not all of which can be exhaustively addressed in a single multiverse analysis. However, follow-up

studies investigating the influence of particular preprocessing steps against reasonable reference pipelines would allow many of them to be tested in a more computationally efficient manner (e.g., ref. 39,), and may provide insight into their impact on decoding performance or feature interpretability.

In summary, the present study systematically varied a wide range of EEG preprocessing steps to evaluate their impact on decoding performance. For EEGNet decoding, high high-pass filter cutoffs and baseline correction increased decoding performance, whereas for time-resolved decoding, high high-pass filter cutoffs, low low-pass filter cutoffs, and linear detrending increased decoding performance. Importantly, all artifact removal steps decreased decoding performance, likely due to covariation between artifacts and experimental condition, effectively reducing the proportion of non-neural features that spuriously enhanced decoding performance. The influence of other preprocessing steps varied depending on the experimental paradigm. We suggest to carefully select EEG preprocessing steps for decoding, as selecting steps that maximize decoding performance is associated with limitations in downstream feature interpretability.

## Materials and Methods
### Data sets
We used the openly available ERP CORE dataset[18,19] (https://osf.io/thsqg/). 40 participants underwent six different EEG experiments to identify the following seven event-related potential (ERP) components: error-related negativity (ERN), lateralized readiness potential (LRP), auditory oddball-related mismatch negativity (MMN), face-related N170, visual search-related N2pc (N2 posterior-contralateral), semantics-related N400, and visual oddball-related P3[18]. ERN and LRP were assessed using the same paradigm analyzed differently. EEG was recorded using 30 scalp electrodes (Fig. S28), along with three electrooculogram (EOG) channels, two positioned lateral to the outer canthus of each eye and one below the right eye[18]. For details of the experimental design and paradigms, we refer to Kappenman et al.[18]. We will only address changes to their processing or aspects that seem important for the purposes of this study or its reproduction. In the remainder of this article, we will refer to these experiments by the names of the corresponding components (e.g., MMN, N170, etc., as shown in Table 1).

### Data preparation
Data preprocessing and modeling was done using a high-performance computing cluster of the Max Planck Computing & Data Facility (Garching, Germany). Each experiment was preprocessed using *MNE* (v. 1.6.1)[57] for Python (v. 3.11.6). Triggers from the raw signal were pruned to retain only relevant events (Table 1). Trials of ERN and LRP were analyzed relative to button press, while all other experiments were analyzed relative to stimulus onset (Table 1). Contrary to Kappenman et al.[18], we used a pre-stimulus instead of a pre-response baseline window in the ERN experiment to minimize systematic differences between conditions in the baseline period[41].

Decoding was performed using data from all available scalp electrodes, in contrast to ERP analysis, where it is common practice to analyze data from individual electrodes. Hence, conditions relating to hemisphere, i.e., contralateral & ipsilateral hemisphere, cannot be meaningfully integrated into a whole-brain decoding logic. For this reason, we assigned the trials of some experiments into new conditions for decoding. For LRP decoding, trials were sorted into left and right responses. For N2pc decoding, we sorted trials into left and right targets (Table 1).

Stimulus event times were shifted forward by 26 ms to account for the delay of the LCD monitor[18]. Data were downsampled from 1024 Hz to 256 Hz. The raw data channels remained in single-ended mode, so no reference was yet applied. The signal from the two horizontal EOG channels was subtracted to form a single horizontal EOG channel[58]. Similarly, the signal from Fp2 was subtracted from the vertical EOG channel, positioned below the right eye, to form a single vertical EOG channel[58]. The data were then temporarily re-referenced to Cz. The final reference electrode was later

determined by the respective forking path. These data then entered the multiverse preprocessing.

### Multiverse preprocessing
All preprocessing steps were restricted to functions included or closely related to the *MNE* package. Guided by preprocessing steps that were investigated by previous EEG multiverse-like studies[12,13,39], we varied ocular artifact correction using ICA, muscle artifact correction using ICA, LPF, HPF, re-referencing, detrending, baseline correction, and artifact correction using autoreject[16,17]. Figure 1 provides an overview of the different preprocessing steps. A total of 2592 forking paths resulted from the systematic variation of the chosen preprocessing steps, i.e., the Cartesian product of all preprocessing steps' options. We chose the same preprocessing steps across all datasets, although the literature suggests a different range of preferred settings depending on the shape of the component. Furthermore, we only included steps that did not require manual intervention by an analyst. Manual intervention is practically not feasible in the multiverse setting and is less consistent, therefore limiting reproducibility[59].

Theoretically, one could extend the multiverse by additional preprocessing options, such as different methods for ocular artifact correction. However, we decided to limit the number of options for all preprocessing steps to keep the computation time within feasible limits. Furthermore, we did not systematically alter the order of the different preprocessing steps, as this would have led to a combinatorial explosion of forking paths. The selected order roughly followed two previous studies[12,13], however, with ocular and muscle artifact corrections positioned at the beginning to avoid contamination of other preprocessing steps.

For most preprocessing steps, but also for later decoding analysis, we deliberately used the default (hyper-) parameters, if not specified otherwise. Furthermore, we stayed as close as possible at the corresponding tutorials provided by the software packages deployed. In the following passages, we briefly describe the individual preprocessing steps and their variations in the multiverse.

**Ocular artifact correction**. In forking paths, including ocular artifact correction, threshold-based ICA was performed using the corresponding *MNE* function. First, the raw signal was copied and a 1 Hz[60] HPF was applied to the copy (see Table S6 for filter characteristics). Then an ICA was performed across EEG channels using the *picard* method[61], setting the maximum number of iterations to 500 and estimating 20 components. The *find_bads_eog* function with default parameters was used to correlate each component with the artificially generated EOG channels. Components were classified as ocular artifact if passing a threshold based on adaptive $z$-scoring. The ICA solution was then applied to the unfiltered data, effectively subtracting the artifact components from the raw signal. It should be noted that recent studies have found evidence, that using a 1 Hz HPF potentially misses parts of the artifactual signal situated below 1 Hz, leading to imperfect artifact removal[62], which could in turn either decrease decoding performance due to remaining noise, or even lead to increased decoding performance compared to alternative approaches if the residual artifact was systematically associated with the experimental condition. Similar to other processing steps, we closely followed the respective *MNE* tutorials when implementing ICA. However, there exist more sophisticated approaches that exploit, for instance, pretrained classifiers for component categorization[56].

**Muscle artifact correction**. A similar procedure was deployed to that used for ocular artifact correction, using the *MNE* function *find_bads_muscle*[63,64]. In short, three criteria were applied to determine if a component represented a muscle artifact[63,64]: spectral slope, peripherality, and spatial smoothness. Because the log-log spectral slope was measured in the range of 7 Hz to 45 Hz, muscle artifact correction was implemented before the LPF. As with ocular artifact correction, the ICA was estimated on a copy of the data filtered at 1 Hz (see Table S6 for filter

characteristics). The ICA solution was then applied to the unfiltered data to remove artifactual components.

**Filtering.** The default filter function of *MNE* was applied to all channels of the raw time series, using a linear FIR filter with cutoffs defined in the respective forking paths (Fig. 1). The filters were one-pass, zero-phase, non-causal band-pass filters (in forking paths with both cutoffs, otherwise HPF or LPF) with a windowed time domain design method (*firwin*), and a Hamming window with 0.0194 passband ripple and 53 dB stopband attenuation. The lower and upper passband edges were defined by the HPF and LPF cutoffs of the respective forking path. Other characteristics were automatically determined by *MNE* using the cutoffs, and are illustrated in Table S6 for the different forking paths. 45 Hz was defined as the upper limit for the LPF, since the AC of most countries is typically transmitted at a frequency of 50 Hz to 60 Hz. The LPF of 6 Hz was included to exclude alpha activity[33]. The HPF cutoffs were motivated by a range commonly used in EEG literature.

**Referencing.** The channels of the raw data were re-referenced to either a single channel (Cz), the average of two channels near the mastoids (P9/P10), or the average of all channels (Fig. 1). According to previous studies, the average of P9 and P10 provided cleaner signals than the commonly used mastoids[18]. Cz or nearby FCz are often used both as an online reference and as a reference during analysis. For convenience, since some of the ERPs of the currently analyzed experiments use FCz as the channel of interest (Table 1), we deployed Cz as a reference option in the multiverse. Finally, the average reference is often used to cancel common-mode noise and to have a more intuitive interpretation of the topographic distributions of the signals.

**Epoching.** Epochs were created for a time window of 1.2 s for each experiment (Table S5). This interval included the same interval as the original study[18]. To be able to increase the baseline period in the multiverse, we added 0.2 s at the end for ERN and LRP, or added 0.2 s at the beginning for all other experiments (Table S5).

**Detrending.** In some forking paths, linear detrending was performed prior to baseline correction. When applied, a linear function with intercept and slope was fitted to each epoch and channel using least squares. The signal was detrended by retaining only the corresponding residuals.

**Baseline correction.** Baseline correction was applied in a subset of forking paths by first averaging the predefined baseline window in each channel and epoch. The average was then subtracted from the epoch for the respective channel and epoch. The baseline period varied between forking paths (Table S5). If applied, baseline correction was either set to the same 200 ms period for each experiment as in Kappenman et al.[18] (but not for ERN), or extended to 400 ms. Crucially, for some response-locked ERPs such as the ERN, a response-locked baseline window is not optimal, since the conditions (error vs. correct) can already be represented in pre-response components extending into the baseline[41]. Further, systematic differences in reaction time between the two conditions can lead to overlap with different segments of the preceding stimulus ERPs[41]. For this reason, we adjusted the baseline windows to precede the stimulus instead of the response in the ERN experiment (Table S5). Finally, the baseline windows were chosen to end at the same time point within each experiment, resulting in identical post-baseline interval lengths across forking paths (Table S5).

**Artifact correction using autoreject.** For further artifact correction on the epochs, the *autoreject* package (v. 0.4.3) was used in a subset of forking paths[16,17] (Fig. 1). In short, the package automatically detects and either rejects or interpolates bad sensors and bad trials based on their peak-to-peak voltages. The rejection thresholds were determined by 5-fold cross-validation. Here we used autoreject in two versions. In the *interpolate* version, the autoreject hyperparameters were set to values such that all bad sensors were interpolated rather than rejected, ensuring that the resulting number of trials for these forking paths remained identical. The advantage of this approach is, that all trials are kept for later analysis. In the *reject* version, we provided autoreject with reasonable hyperparameters for *consensus* (0.2, 0.4, 0.6, 0.8) and *n_interpolate* (4, 8, 16, 32). A grid search was performed to find optimal values[16,17]. If only a few electrodes exceeded the estimated threshold, they were interpolated by neighboring electrodes. If many electrodes exceeded the threshold, the entire epoch was discarded. We used 25% of the epochs to fit the autoreject model.

While the previous preprocessing steps did not change the number of epochs per condition, the second autoreject version could drastically reduce the number of epochs. If a forking path had too few epochs per condition for later cross-validation, the forking path of that participant was discarded. This happened only in rare cases. Note that autoreject shows substantial variability in the number of dropped epochs with different random seeds (Fig. S29 & S30, Supplementary Text – Variability due to random seed in autoreject).

## Evoked responses

We computed exemplary grand average evoked responses as a proof of principle to ensure that data processing worked as expected. Therefore, for each experiment, we chose a forking path that was related to the preprocessing reported in the original paper, releasing the dataset[18]. That is, for all experiments except N170, re-referencing was done on mastoids (average of P9 & P10), with an HPF of 0.1 Hz, and no LPF applied. ICA was used for both ocular and muscle artifact correction, as well as autoreject in the interpolated version. We chose here to use all artifact correction steps in this example forking path, as the original study performed multiple, including manual – artifact correction steps. The baseline period was 200 ms, and no detrending was performed. For N170, re-referencing was done using an average reference. The remaining steps were the same as for the other experiments (Fig. 1).

For LRP and N2pc, channels were combined to obtain signals from channels contralateral and ipsilateral to the target (N2pc) or response (LRP) (Table 1). This aggregation was performed only for visualization of evoked responses in accordance with Kappenman et al.[18], but not for decoding.

## Decoding models

We selected a neural network-based and a time-resolved framework for decoding. The former is more commonly used in BCI research[20], whereas the latter is often used in basic research on the time course of category discrimination[33,65–70].

**Trial-wise decoding using EEGNet.** The neural network-based decoding model employed was EEGNet (v. 4)[20], implemented in the *braindecode* toolbox (v. 0.81)[44,57]. For each forking path, the preprocessed trials were rescaled using exponential moving standardization. The MMN, P3, and ERN experiments had highly imbalanced classes, while the LRP experiment was only slightly imbalanced. Furthermore, autoreject introduced an imbalance when used in the reject version. Therefore, class weights were computed separately for each forking path, experiment, and participant.

Next, the order of the trials was shuffled, and a stratified 5-fold cross-validation split was defined. The model was initialized with its default hyperparameters, a batch size of 16, an exponential linear unit activation function, a dropout rate of 0.25, a learning rate of 0.01, a stochastic gradient descent optimizer, and a negative log-likelihood loss function. The maximum number of epochs (i.e., training cycles, not electrophysiological epochs) was set to 200. See Table S7 for a comprehensive list of hyperparameters, and Table S8 for the layer structure. Scoring was performed using balanced accuracy for the 2-class case[71]. Balanced accuracy is the mean

of sensitivity and specificity, and is equivalent to accuracy for class-balanced data. For each cross-validation split, the model was fitted to $\frac{4}{5}$ of the data and evaluated on the remaining $\frac{1}{5}$ of the data. The resulting test accuracies across the 5 splits were averaged. The approach was repeated for each forking path, experiment, and participant.

**Time-resolved decoding using logistic regression**. In the literature, raw electrode voltages[33,68] or averaged pseudo-evoked potentials[69,70] across trials, but for each time point separately, are used to train classifiers (Fig. S31). In the present study, we used the raw trials for classification. Data of each participant, time point, and channel were standardized individually by removing the mean and scaling to unit variance across trials. As with EEGNet, balanced accuracy was used for scoring, and class weights were computed individually for each forking path, experiment, and participant. A logistic regression estimator (*sklearn* v. 1.3.1) was used with a *liblinear* solver[72], *L2* penalty, C = 1, tolerance of 0.0001, and a maximum number of iterations of 100, applied to each time point with the *SlidingEstimator* function within *MNE*. See Table S9 for a comprehensive list of hyperparameters. Decoding windows corresponded to the entire trial length (including baseline) and were kept equal in length across experiments, but differed in onset and offset (Table S5). A stratified 10-fold cross-validation was performed, and the test accuracies were averaged across folds, resulting in one decoding time series for each forking path, experiment, and participant.

**The issue of data leakage in EEG decoding**. A topic that is closely monitored in the machine learning community, but often neglected in the neuroscience community, is data leakage[73,74]. Through data leakage, information from the test data is introduced into the training data[75], leading to an overestimation of decoding performance. Recent studies have investigated how leakage is introduced when decoding is performed across participants with segment-based instead of subject-based holdout sets during cross-validation[73]. However, we are in the context of within-participant decoding. Others have investigated the leakage of test trials into the training data or the leakage of test data into the estimation of data features[76]. In our analyses, we used k-fold cross-validations separating training and test epochs. However, the data were jointly preprocessed, leading to imperfect separation. First, the entire time series is used to estimate the ICA solution, meaning segments later assigned to the test set contribute to the removal of components from both the test and training sets. Second, when applying the HPF, test trials can influence training trials due to proximity in the raw time series. Lastly, autoreject involves both training and test data to jointly estimate hyperparameters and rejection thresholds. While this joint preprocessing does not directly resemble the estimation of predictive features for modeling, there could theoretically still be a nuanced influence between data segments, which we name *latent leakage*. This leakage could systematically lead to higher decoding accuracies for forking paths, including the affected preprocessing steps. To the best of our knowledge, the potential of latent leakage has not been systematically analyzed and tends to be ignored in EEG studies.

One possible countermeasure for latent leakage would be to split the time series into a fixed number of temporally separated segments, the trials of which are then used in different cross-validation folds. Preprocessing operations should then be applied only within each segment. However, many algorithms (e.g., ICA or autoreject) may be less stable when run on less data. A different approach could be a separate preprocessing for each cross-validation fold, so that the training data (e.g., 80%) is used to estimate the independent components and autoreject thresholds, and these models are then applied to the test data without separate estimation. However, such a procedure would increase the computation time enormously and is computationally not feasible in the multiverse. Both approaches would also work more straightforward if precautionary measures in paradigm designs were ensured, such as, for instance, an equal distribution of conditions in the different temporally separated segments.

To preclude noticeable latent leakage in our data, we analyzed whether perfect separation of train and test data decreases decoding performance by manipulating HPF, ocular ICA, and autoreject in a subset of forking paths to achieve perfect separation (see Supplementary Text - Influence of latent leakage on decoding performance). The results showed no increase in decoding performance, i.e., no influence of latent leakage (Fig. S32). However, we emphasize that latent leakage should be further investigated in the future in the context of different preprocessing steps outside the current multiverse.

## Quantification of decoding performance

In the current article, we report only test accuracies, i.e., we use the terms accuracy and decoding accuracy synonymously to test accuracy, as an abbreviation for averaged, cross-validated, balanced test accuracy. We quantify decoding performance either by test accuracy (EEGNet) or by the *T*-sum of the calculated group statistics (time-resolved).

**EEGNet**. For trial-wise decoding using EEGNet, we directly used the averaged, cross-validated, balanced test accuracies. This resulted in up to 2592 test accuracies for each of the 40 participants and each of the 7 experiments. We directly applied a linear mixed model (LMM) (see below), using these test accuracies as the dependent variable.

**Time-resolved**. For time-resolved decoding using logistic regression, we first quantified significant clusters at the group level (i.e., across 40 participants) for each forking path and experiment. This was done using a one-tailed, one-sample permutation cluster mass test[23,77,78]. Briefly, we first defined temporal clusters using a cluster-forming threshold of $p <$ 0.05 separately at each time point ($256 \frac{1}{s}$) on the zero-centered averaged decoding time series. For each cluster – i.e., union of all neighboring time points above threshold – the sum of *T*-values (*T*-sum) was calculated.

Each participant's individual decoding accuracy time series was then randomly multiplied by either 1 or −1, averaged, and zero-centered. Clusters were defined in these permuted datasets using the same cluster-forming threshold. By repeating the permutations 1024 times, we achieved a Monte Carlo sampling of the permutation distribution. The cluster with the highest *T*-sum per iteration was saved for each sample of the permutation distribution. The *p*-value was calculated by relating the *T*-sums of each cluster of the original dataset to those of the sampled permutations, and only clusters falling in the highest 5% quantile were retained.

For each forking path and experiment, this procedure corrects the alpha error of a cluster being significant at the family-wise error rate[77]. The *T*-sums over all significant clusters were then entered as the dependent variable into one linear model (LM) per experiment (see below).

As an alternative approach to quantifying time-resolved decoding performance, we averaged the cross-validated test accuracy values across time points after the baseline, for each forking path, experiment, and participant, and entered these values into one LMM per experiment, similar to the procedure applied to EEGNet accuracies. The results are reported in the Supplements (Fig. S1 & S2). The rationale was that this approach uses accuracy estimates bounded by the interval between 0.5 and 1.0, which are more readily interpretable than *T*-sums. However, a typical study would likely rather be focused on delineating and interpreting significant clusters for a group. Therefore, the *T*-sum approach might be practically more relevant and was chosen to be followed in the main manuscript. Estimation of *T*-sums on the individual participant level would also be possible[77], but was avoided for computational reasons in the multiverse context.

Further, we explored the use of threshold-free cluster enhancement[78] (*TFCE*) instead of defining a fixed cluster-forming threshold. We summed the individual (time point-wise) *TFCE*-values to *TFCE*-sums instead of *T*-values to *T*-sums. Because *TFCE*-sums and *T*-sums correlated strongly across forking paths per experiment (Pearson correlation, all r > 0.96, Fig. S33), we anticipate similar conclusions from downstream analyses and only continued with the approach using a fixed cluster-forming threshold and corresponding *T*-sums.

### Modeling the effects of preprocessing steps

**EEGNet**. We constructed one LMM for each experiment separately. The test accuracies per participant and forking path in that experiment served as the dependent variable. All preprocessing steps were modeled as factors. Each step's version with the smallest intervention served as the reference level (upper option of each step in Fig. 1). We also allowed for two-way interactions between all steps. To account for different levels of decoding performance per participant, a random intercept term for the participant was added. To further account for participant-dependent variability within each factor, and to avoid inflation of the alpha error[79,80], we added random slopes for all main effects and interactions. All LMMs were estimated using *MixedModels* (v. 4.23.1) for *Julia* (v. 1.10.2[81]) and transferred to $R$[82] for further processing using *RCall* (v. 0.14.1) and *JellyMe4* (v. 1.2.1), and the $R$ package *afex* (v. 1.3.1[83]).

**Time-resolved**. After identifying significant clusters in the decoding time series of each experiment and forking path, we calculated the total $T$-sum across all significant clusters for each experiment and forking path as a proxy for decoding performance. We then constructed one LM for each experiment separately with the $T$-sum as dependent variable. All single preprocessing steps were modeled as factors, resulting in a full factorial design. As reference-level for each preprocessing step served the variation with the smallest intervention (upper row in Fig. 1). Two-way interactions were added between all terms. LMs were estimated using the *stats::lm* function in base $R$[82] (v. 4.2.2).

**Modeling interactions**. Since preprocessing steps build on each other, they are likely to interact. The recommended way to include interactions is to define the most complete model possible[79]. On the other hand, more model terms make it more difficult to estimate and ultimately interpret the resulting model. We tested whether the addition of interaction terms added value by first testing whether a considerably higher proportion of variance ($R^2$) was explained, and second, whether the Akaike information criterion (AIC) decreased. Figure S34 shows that a considerable amount of variance is already explained by the main effects when modeling EEGNet decoding performance, but less so for time-resolved decoding performance. Especially for time-resolved decoding performance, interaction terms added substantial value in terms of explained variance (Fig. S34). More importantly, including interactions always decreased AIC values (Fig. S34).

To keep the model computationally traceable and the results interpretable, we decided for two limitations regarding model interactions. First, we included only two-way but not higher-order interactions, as the number of model terms (both fixed and random effects) increased drastically. Second, we restricted the model to not estimate the correlations between random effect terms, which significantly shortened the computation time.

**Estimation of marginal means**. Marginal means and contrasts were estimated using the *emmeans* package (v. 1.10.0)[84] in $R$. Briefly, emmeans extracts the mean responses from statistical models, and returns average response variables – in our case, model decoding performance operationalized via accuracy or $T$-sum – for each level of a categorical predictor variable. First, we estimated marginal means for all main (fixed) effects to interpret the influence of each preprocessing choice on model decoding performance. Second, we also estimated marginal means for each interaction effect. This step yielded marginal means for each factor level of a given preprocessing choice, grouped into the factor levels of another preprocessing choice. Similarly, contrasts were computed between all pairs of levels.

Statistical significance was calculated model-wise, i.e., per experiment and decoding framework. On the marginal means of each LMM and LM, we applied omnibus $F$-tests to each preprocessing step (main effects and interactions) to determine a significant contribution of the preprocessing step to the decoding accuracy, without further correction for multiple comparisons. Then, pairwise $T$-ratios and $p$-values were computed for the main effects between the factor levels of each preprocessing step using Tukey adjustment within each preprocessing step[85].

**Evaluating the impact of manipulating a single preprocessing step**. In an additional analysis, we analyzed the impact of changing a single preprocessing step on decoding performance. For this, we used the example forking path (Fig. 1) as a reference. This forking path was defined as a reference with the same rational than for the ERP visualization, because it is loosely related to the preprocessing pipeline used in the accompanying article released with the dataset[18]. We then systematically tested how decoding performance changes when manipulating one single processing step, but leaving all others unchanged.

**Impact of preprocessing on baseline artifacts**. We analyzed which preprocessing steps increased decoding performance during the baseline period, i.e., the probability for baseline artifacts. First, we defined the presence of a baseline artifact when significant clusters occurred within or across the baseline windows (Table S5), e.g., for N170, clusters extending to a time point ≤ 0.0 s. For ERN, the baseline period was not tied to the response onset and may therefore be outside the decoding range (Table S5). Therefore, clusters in ERN emerging before the 0.4 s pre-response time point were defined as baseline artifacts, similar to LRP (Table S5).

To assess the contribution of each preprocessing step to the presence of a baseline artifact, we modeled the presence of an artifact in a forking path as a function of all preprocessing steps separately for each experiment. Binary logistic regressions were applied using maximum likelihood estimation with default optimization settings (i.e., convergence threshold of $10^{-8}$ and a maximum of 25 iterations). We computed likelihood ratio tests to obtain $\chi^2$ statistics and $p$-values, indicating if the exclusion of a preprocessing step significantly reduced model fit ($\alpha = 0.01$).

**Treatment of statistical significance**. We deliberately try not to emphasize statistical significance on the multiverse outputs in the remainder of the manuscript. First, by its very nature, the multiverse could theoretically be inflated by adding more preprocessing choices, either entire steps or variations of steps. If we assume that we add another step with two options, and for simplicity, assume no interaction with the previous steps, we effectively doubled the number of forking paths for each group. Thus, small effects could be driven to significance by arbitrarily inflating the multiverse. Second, we used both LMMs (EEGNet) and LMs (time-resolved) to infer the influence of preprocessing steps. Due to the random effects in LMMs and the larger number of data points entering the model (factor 40), both parameter estimation and the diagnostic performance of the statistical analyses could vary greatly between the two approaches. Third, the dependent variables were different in LMMs and LMs. A reader might be tempted to compare the significances of processing steps between EEGNet and time-resolved approaches, which is not possible for the present analyses. For these reasons, we highlight steps that appear important by analyzing the charts rather than only $p$-values.

### Reporting summary

Further information on research design is available in the Nature Portfolio Reporting Summary linked to this article.

## Data availability

Large files are shared via Zenodo (https://zenodo.org/records/14223514), including test accuracies or $T$-sum values per participant, forking path, experiment, and time point. To allow for a simple comparison of individual preprocessing pipelines and their impact on decoding performance, we deployed an online dashboard at https://multiverse.streamlit.app. The raw dataset used in our analyses is shared by its authors[18,19] at https://osf.io/thsqg/.

## Code availability

A GitHub repository containing all scripts is available at https://github.com/kesslerr/m4d, including *Bash* & *Python* scripts handling the complete multiverse preprocessing and decoding on a high-performance computing cluster. The repository also comprises all analysis scripts to compute LMMs in *Julia*, and to perform all remaining analyses in *R* based on a *targets* pipeline[86]. Comprehensive lists of package versions in all used programming languages are available at https://github.com/kesslerr/m4d/tree/main/env.

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

## Acknowledgements

This work was supported by the German Research Foundation, DFG Heisenberg Program Grant 433758790 (MAS) and the Jacobs Foundation, Research Fellowship (MAS).

## Author contributions

Conceptualization: R.K., A.E. Formal analysis: R.K. Funding acquisition: M.A.S. Investigation: R.K. Methodology: R.K., A.E. Project administration: R.K. Resources: R.K., M.A.S. Software: R.K. Visualization: R.K. Writing – original draft: R.K. Writing – review & editing: R.K., A.E., M.A.S.

## Funding

## Competing interests

The authors declare no competing interests.
