## [Transparent Peer Review file · Communications Biology]

How EEG preprocessing shapes decoding performance

Corresponding Author: Dr Roman Kessler

Version 0:

Reviewer comments:

Reviewer #1

(Remarks to the Author)

The authors provide a robust and comprehensive analysis of how EEG preprocessing influences decoding performance, using a systematic multiverse approach to evaluate multiple preprocessing pipelines. The study is well-structured, methodologically rigorous, and provides valuable insights for the field of EEG-based machine learning. However, some comments to enhance clarity and completeness are provided below:

Major Comments

- The authors acknowledge data leakage as a limitation, which is appreciated. However, could they test the extent to which it affects classification accuracy? For instance, could they compare results when preprocessing is applied within the training set only versus the entire dataset? This would strengthen confidence in their conclusions and align with best practices in ML validation.
- Could the authors provide additional details on the architectures and hyperparameters of EEGNet and the time-resolved classifier? Specifically, information on layer structures, activation functions, dropout rates, batch size, learning rate, and training epochs would improve reproducibility. Providing a clear architecture table or a link to code would significantly improve reproducibility.
- Figure S28, S29 demonstrate that autoreject removes different epochs depending on the random seed, introducing stochastic variability in classification results. Could the authors discuss whether results are stable across multiple runs with different seeds? A sensitivity analysis showing whether classification accuracy remains consistent would be valuable.

Minor Comments:

- Would the results reported for decoder performance be generalisable to other datasets? Could authors do a sub-analysis to test this, or if this is beyond the scope of the current study, the authors could discuss potential generalisability issues.
- To improve clarity, could the authors provide a specific example within the citation (21-23) to illustrate the referenced concept?
- To better capture classifier performance beyond accuracy, could the authors include additional metrics such as recall, specificity, AUC, and precision? This would be particularly useful for understanding class imbalances and systematic errors in decoding.
- To provide a clearer reference point, could the authors include a comparison of decoding performance on unprocessed EEG data versus pre-processed data? This would help quantify the extent to which preprocessing enhances or alters classification outcomes.
- Could the authors investigate whether preprocessing choices lead to spurious decoding in baseline periods? This could help determine whether classifiers are relying on residual artifacts rather than meaningful neural signals.
- Could the authors discuss the trade-offs between maximizing decoding accuracy and maintaining neurophysiological interpretability? This would be particularly useful for ERP researchers who prioritize signal clarity over classification performance.
- Could the authors highlight which preprocessing choices are most likely to introduce baseline artifacts (Figure S19)? This would help researchers avoid overestimating classification accuracy due to preprocessing-induced distortions.

Reviewer #2

(Remarks to the Author)

The paper “How EEG preprocessing shapes decoding performance” explores how different preprocessing steps impact EEG classification using EEGNet and time-resolved logistic regression. By testing multiple preprocessing pipelines across seven ERP experiments, the study shows that choices like filtering, artifact correction, and baseline correction can significantly affect decoding accuracy. Using linear mixed models (LMMs) on T-sum for time-resolved analysis, the authors provide a clear and systematic way to measure these effects. Their approach is rigorous, transparent, and valuable for improving EEG decoding methods, offering practical insights for both research and brain-computer interface applications.

The study provides important insights into EEG preprocessing but is limited by its narrow range of artifact rejection methods, which do not reflect common practices in the field. Many widely used techniques, such as ASR, ICLabel, MARA, and the PREP pipeline, are available only in MATLAB and were not included or replicated in Python, despite MNE being used in only ~5% of EEG studies as of 2025. This omission weakens the generalizability of the findings. Expanding the preprocessing pipeline to include these methods would make the results more representative of real-world EEG research. Other limitations are indicated below.

Limitations and Suggested Improvements

- 1. Limited Artifact Rejection Methods.** The study relies on MNE for artifact rejection, which supports only a small subset of existing methods. Many widely used approaches, such as Clean_rawdata, Cleanline, ASR, ICLabel, MARA, and the PREP pipeline, are primarily implemented in MATLAB and are not available in MNE. Given that MNE is used in only about 5% of EEG publications as of 2025, preprocessing the data in MATLAB or implementing Python equivalents for these methods would have provided a more comprehensive analysis of artifact rejection.
- 2. LMM Significance and Figure 5 Improvements.** Linear Mixed Models (LMMs) inherently control for multiple comparisons, so additional corrections are unnecessary. Figure 5 should highlight only significant differences rather than showing all variations in color. Additionally, a row grouping all experiments together would help visualize overall trends. The rationale behind integrating all experiments into a single LMM model needs clearer explanation in the methods section.
- 3. ICA Application and Artifact Detection.** The application of Independent Component Analysis (ICA) in the default pipeline is suboptimal. Standard EEG preprocessing applies ICA after filtering, which was only tested in an alternate multiverse but should likely be the default. Furthermore, artifact detection based on thresholding activity is simplistic and not used in practice—modern EEG pipelines use machine learning-based ICA selection methods like ICLabel. To reflect this limitation, the study should rename ICA as “Naive ICA” to clarify that it does not represent the current state-of-the-art approach. This point should also be mentioned in the discussion.
- 4. Potential Overfitting in EEGNet Training.** EEGNet is trained with a fixed 200 epochs, but no evidence is provided to confirm that this choice prevents overfitting. Some validation data should be presented to justify this parameter. Additionally, other EEGNet training parameters are not disclosed, making it difficult to replicate the results. The version of BrainDecode used should also be specified.
- 5. In the abstract, it is stated, “All artifact correction steps reduced decoding performance across all experiments and models, while higher high-pass filter cutoffs.”** This conclusion aligns with that of reference 33, “Delorme, A. EEG is better left alone. *Sci. Rep.* 13, 2372 (2023).” It is important to emphasize this point in the discussion, particularly given that this paper incorporated a broader range of artifact rejection methods and EEG references, suggesting that the current approach might generalize to other rejection methods.
- 6. Software Reproducibility and Filtering Details.** The study does not document the versions of MNE, Autoreject, or other key software used, which is critical for reproducibility. Additionally, filter characteristics (e.g., filter order, roll-off, and type) should be explicitly stated to ensure precise replication.
- 7. Cross-Validation and Confidence Intervals.** The study uses 5-fold cross-validation, but confidence intervals are not provided in Figure 6. Adding confidence intervals would clarify the variability in classification accuracy and improve result interpretation.
- 8. Alternative to T-sum: Threshold-Free Cluster Enhancement (TFCE).** Instead of using T-sum for statistical testing in time-resolved decoding, the study could adopt Threshold-Free Cluster Enhancement (TFCE). TFCE improves sensitivity and avoids arbitrary cluster thresholds, making it a more robust alternative to traditional cluster-based permutation testing (CBPT).
- 9. Availability of Raw EEGNet and Time-Resolved Decoding Results for Further Analysis:** The study presents a rich dataset with extensive decoding results from both EEGNet and time-resolved logistic regression across a large multiverse of preprocessing pipelines. However, the raw classification results are not available. Releasing these raw results would provide a valuable resource for the community.

Reviewer #3

(Remarks to the Author)

This article reports the results of a fantastically detailed, well thought out, and extensive study of the effects of a good number of EEG preprocessing steps on the decoding of event-related potentials from cognitive task-related EEG. The methods used are robust, as are the statistical tests. Conclusions drawn are accurate, although I have suggested a few

additional caveats and considerations. I have also suggested a number of considerations for the authors below. These considerations do not indicate that the study is not already excellent, they are merely suggestions for improvements, provided from my perspective as a researcher experienced with EEG preprocessing and ERP analysis.

Abstract:

- Suggest stating that other pre-processing steps were optimal for specific experiments / event-related potential components (because it's not possible to discern between whether it's the influence of preprocessing on the experiment or on the specific ERP).
- While it is true that all artifact correction steps reduced decoding performance across all experiments and models, I would suggest that a strong caveat be provided that from the current results, it's impossible to tell if this might be the influence of artifacts in the data. Although in BCI applications it may be useful, if decoding studies are implemented to tell us more about neural activity, then higher decoding accuracy in the context of higher artifact influence is counter-productive rather than useful. Sole focus on decoding accuracy is an issue here, in a similar way to studies that emphasize high classification accuracy at the expense of data leakage (and thus less practically useful results). I suggest statements to these effects be provided strongly and clearly in the abstract, so future research does not emphasize high decoding accuracy at the expense of practical usefulness as a result of the influence of this manuscript (the statement "if not corrected, artifacts facilitate decoding but compromise conclusive interpretation" is not strong or clear enough).

Introduction:

- Line 38 – similar to my comments on the introduction, I would caveat these statements with the note that maximizing decoding performance is not useful, even in contexts with compromised data quality, if the maximising comes at the expense of validity or the potential to lead to practical utility. E.g. it would be trivially easy to maximise decoding performance by asking participants to blink on certain trials and not others, but not practically useful.
- I'd suggest expanding a bit on how decoding can be useful (for more than just BCI and reducing the multidimensional space of EEG data).
- Line 63 – suggest briefly explaining how each of the EEGNet and time-resolved logistic regression classifier works for readers who might be less familiar with them. Particularly for the logistic regression classifier, since the reference provided for this doesn't seem to describe the method (Gramfort, A. et al. MNE software for processing MEG and EEG data. *NeuroImage* 86, 446–460 (2014)).

Results:

- Line 162: I think it's worth noting in the results that it isn't clear whether removing EOG (and muscle) artifacts decreases decoding performance because: 1) the artifacts are informative of differences between conditions, or 2) removing the artifacts also removes informative neural activity. Recent research has suggested that removing EOG artifacts with ICA does also remove neural activity due to the imperfect separation of neural and artifact source activity by ICA: Bailey, N. W., Hill, A. T., Godfrey, K., Perera, M. P. N., Rogasch, N. C., Fitzgibbon, B. M., & Fitzgerald, P. B. (2024). EEG is better when cleaning effectively targets artifacts. *bioRxiv*, 2024-06. While this might typically be considered a discussion point, I suspect some readers may not read the entire article, and this point is fairly critical for consideration in future research, so I think it is worth the authors taking steps to ensure it is not missed.
- Line 233: Suggest re-writing the following sentence for readability: "Several effects can be observed when using autoreject in the version, in which noise-contaminated trials were discarded rather than interpolated. When not applying a HPF, linear detrending or baseline correction but using autoreject in this version, decoding accuracies were lower"
- Figure 6 – I think the y-axis should be labelled as "Balanced Accuracy" rather than marginal mean?
- Line 256 – As far as I can tell, the authors haven't specified whether causal or acausal filters were applied. If acausal filters were applied, this could explain some of the baseline period decoding accuracy, with the acausal filter smearing weights to influence activity at earlier timepoints (in contrast to the causal filters where filter weights only influence subsequent timepoints). This issue should be explained in detail, as it likely affects many decoding studies.
- Line 276 – should it be 0.4 seconds rather than 0.4ms pre-response? (0.4ms would be practically the same as just using the response timepoint).

Discussion:

- Line 312 – it is true that artifact correction steps impaired decoding. But stating the result like this, without noting that remaining artifacts in the data reduces the practical utility of the decoding analysis, emphasizes decoding accuracy over usefulness. This is common across machine learning applications, but should be strongly resisted. High accuracy is only worth reporting if it is useful, and lower accuracy that is more informative/useful should be preferred over high accuracy that is only the result of artifact contamination. The situation here is complicated by the lack of ability to determine whether it is the artifact itself driving the increased accuracy, or protection against reducing the neural activity as well as the artifact in the artifact cleaning approach. The fact that both the ICA and the autoreject steps lead to lower decoding accuracy may suggest the artifact itself is inflating accuracies, but without certainty. If the authors are able, it would be good to explore the potential explanations further. If not, they might recommend future research explore the point. Future research might use methods that specifically target eye and muscle artifact activity within the ICA decomposition to enable this (which better protects the neural activity outside of those artifacts), for example the methods explained in: Bailey, N. W., Hill, A. T., Godfrey, K., Perera, M. P. N., Rogasch, N. C., Fitzgibbon, B. M., & Fitzgerald, P. B. (2024). EEG is better when cleaning effectively targets artifacts. *bioRxiv*, 2024-06.
- On this point, perhaps the best combination of preprocessing steps for decoding analyses that are aimed at understanding / decoding neural activity are ranked ~100 to ~300 in Figure S18 – the steps that provide high decoding accuracy, but don't show decoding accuracy where there shouldn't be any (prior to the stimulus presentation, where neural activity cannot be responding to the different conditions). If I were writing the paper, I would emphasize this point.

- Line 316 – I'm glad that the authors have addressed this point. However, as noted in my earlier comments, I think the point needs to be highlighted throughout the manuscript. As it currently reads, the rest of the manuscript seems to emphasize decoding performance over the insights into neural activity that might be offered by decoding.
- Line 321 – Bailey et al., (2024) could be cited to support the point that neural signals of interest are also removed by removing artifacts.
- Line 481 – I think the point about artifacts producing higher decoding performance, but this not necessarily reflecting neural activity needs to be noted here also.

Methods:

- Line 525: Does subtracting the EOG channels from each other adversely affect the ICA performance? From my understanding, this is not a typical approach, and using the same reference for all electrodes included in the ICA is important. If these EOG channels were not included in the ICA (which seems to be the case when the ocular artifact correction is explained later), then this should be explicitly stated.
- Line 560: The work introducing the picard ICA method should be cited.
- Line 557: While high pass filtering data at 1Hz prior to the ICA improves ICA decomposition performance, and the approach is somewhat common in the literature, high pass filtering data prior to the ICA, then applying the ICA weights back to the unfiltered data has been shown to lead to worse blink artifact correction performance, and weaker ERP effect sizes. This is likely because there are EOG artifact signals <1Hz, which are not addressed by applying the 1Hz filtered ICA to the unfiltered data. Evidence of this is provided in the supplementary materials of: Bailey, N. W., Hill, A. T., Biabani, M., Murphy, O. W., Rogasch, N. C., McQueen, B., ... & Fitzgerald, P. B. (2023). RELAX part 2: A fully automated EEG data cleaning algorithm that is applicable to Event-Related-Potentials. *Clinical Neurophysiology*, 149, 202-222. I don't expect the authors to re-analyse all of their data based on this point, but it is worth noting as a limitation, with the suggestion that the decoding performance after ICA to correct EOG may have been reduced because of this issue.
- Line 601: Subtraction baseline corrections are likely not optimal for ERPs, as they transpose topographical effects from the baseline period into the active period, and do not actually provide the control for baseline shifts that they are intended to achieve. See the following study for details: Alday PM. How much baseline correction do we need in ERP research? Extended GLM model can replace baseline correction while lifting its limits. *Psychophysiology*. 2019;56(12):e13451. Also see the following study for evidence that this can produce both false positives and false negatives in the ERP active window of interest in real data: Hill, A. T., Chung, S. W., Emonson, M., Corcoran, A. W., Fitzgibbon, B. M., Fitzgerald, P. B., & Bailey, N. W. (2024). Neural differences in conflict monitoring and stimulus expectancy processes in experienced meditators are likely driven by enhanced attention. *bioRxiv*, 2024-12. Similar to the issue with 1Hz high pass filtering prior to ICA, I do not expect the authors to re-analyse their data based on this issue, but it should be noted, with suggestion that future research explore regression baseline correction methods with decoding analyses to see if outcomes are improved over subtraction baseline correction methods.
- Line 676: With the statement: "Data of each time point were standardized by removing the mean and scaling to unit variance." – please indicate whether the scaling is applied per timepoint, across timepoints, including just each individual separately, or all individuals, etc.
- Line 708: I mostly agree with the sentiment in the point about data leakage. However, it will certainly be worth noting that some preprocessing conditions include more potential data leakage than others, so if data leakage has affected decoding performance, then the comparisons between those conditions and conditions that contain fewer steps that might introduce data leakage might be biased.

Supplementary Materials:

- Figure S3 – should there be numbers in this figure like in the preceding figure? I'd suggest explaining why the boxes are greyed out also (I assume because they reflect matches to the reference pipeline?).
- Figure S4 – great figure, but I'd suggest expanding the y axis so that A and B separately take up the whole page, so that readers can more easily see which steps are included in the top ranked pipelines.
- Figure S5 – it's surprising that simply subtracting Cz from all the signals inflates decoding performance so much compared to average or P9/10 references. This suggests that the Cz signal itself is highly informative. Perhaps this should be noted in the discussion as a point for further exploration in future research.
- Figure S6 onwards – I think the authors should replace "see figure 6 for details" with an explanation suited to each figure, to make it easier for the reader to understand each figure in its own context (rather than having to refer between the main manuscript and the supplementary materials to understand each supplementary materials figure). Also, I think the y-axis are mislabelled for at least some of these figures.
- Figure S19 - Even if very few forking paths produced baseline artifacts, if the goal is to decode conditions based on neural activity, any baseline artifact at all suggests that goal is not being achieved, and performance is suspect to contamination by non-neural sources. As such, even for the later ERPs (N400 and P3) which should show low baseline decoding artifact, removing artifacts with ICA seems like a critical step. The issue of decoding performance produced by non-neural sources should be considered more critical than low decoder performance. I view this as analogous to data leakage – high performance doesn't matter if performance is due to an artifactual signal, regardless of whether the artifact is in the classification algorithm design or the signal processing pipeline.
- Figure S25 – Based on the description of this figure early in the supplementary materials, I think it would be worth noting in the results and discussion the point that a lower low-pass filter might replace muscle artifact rejection to enable higher decoding performance (which also eliminates the concern that the muscle artifact itself is leading to higher decoding performance).
- Figure S25 – The figure caption should explain what this alternative multiverse reflects (the different ordering of preprocessing steps?)
- Figure S29 – is this an example from just one participant? I think the caption could explain more clearly.

- Figure S31 – the title of one of the panels in the top part of the figure seems to be wrong (given the values in the two panels are reverse polarity from each other).
- Table S1 – typo – “significance”.

Version 1:

Reviewer comments:

Reviewer #1

(Remarks to the Author)

Thank you to the authors for their detailed and thoughtful revisions to both the manuscript and supplementary material. I commend the authors for their comprehensive and constructive responses to all reviewer comments. The revised manuscript has been significantly improved in clarity, methodological transparency, and contextualisation of its findings.

The authors have satisfactorily addressed my previous concerns by:

- Clarifying the methodological framework and underlying assumptions;
- Providing additional context on model generalisability and interpretability; and
- Strengthening the limitations section to reflect the preliminary and exploratory nature of the work.

While some aspects remain exploratory due to inherent constraints, the manuscript is transparent about these issues, and I believe it will positively influence future research in the field.

I find the study to be methodologically sound, statistically appropriate, and reproducible based on the information provided. I therefore support publication of the manuscript in its current form.

Reviewer #2

(Remarks to the Author)

Regarding my earlier comment on the Limited Artifact Rejection Methods, I do not believe it was adequately addressed in the rebuttal. While I understand the computational constraints involved in expanding the pipeline to include more artifact rejection methods, this is not a sufficient justification for omitting the most widely used techniques in the field—such as `clean_rawdata`, `ICLabel`, `MARA`, and `PREP`. These methods are integral to current EEG preprocessing practice, and excluding them limits the generalizability of the findings.

Given this methodological restriction, the current title “How EEG preprocessing shapes decoding performance” overstates the scope of the study. To more accurately reflect the specific implementation context, I strongly recommend revising the title to something similar to

“How MNE-based EEG preprocessing shapes decoding performance.”

This change would provide appropriate transparency about the methodological boundaries and prevent misleading readers regarding the study’s generalizability.

That said, I appreciate that my other comments were addressed thoroughly and constructively in the revised manuscript.

Reviewer #3

(Remarks to the Author)

Thank you for addressing my previous comments, I think the manuscript is improved by the changes, and am satisfied that it now accurately presents the results and conclusions from this very interesting study.

Minor points remaining:

Page 17 and 20: the explanation on page 20 that copying the 1Hz high pass filtered ICA decomposition to the raw data or data with a lower high pass filter setting may lead to poorer artifact reduction is not explained very clearly, and I cannot see where the adjustment has been made on page 17 to address this point.

Page 12 and other points: I’m not sure “facilitated” is the best word when describing an increase in baseline artifact. Perhaps “increased the number of pipelines that showed a baseline artifact” or “increased the number of timepoints that showed a baseline artifact” or “increased decoding performance during the baseline period, suggesting a baseline artifact” would be better?

Response to reviewers

We sincerely thank all three reviewers for their thoughtful and constructive feedback. We have carefully reviewed all comments and incorporated the suggested revisions wherever feasible. For suggestions we were unable to implement, we have provided clear justifications. Below, we present a point-by-point response, with our replies highlighted in blue. Our provided line numbers correspond to the *manuscript version with track changes*.

Reviewer #1

The authors provide a robust and comprehensive analysis of how EEG preprocessing influences decoding performance, using a systematic multiverse approach to evaluate multiple preprocessing pipelines. The study is well-structured, methodologically rigorous, and provides valuable insights for the field of EEG-based machine learning.

However, some comments to enhance clarity and completeness are provided below:

Major Comments

- The authors acknowledge data leakage as a limitation, which is appreciated. However, could they test the extent to which it affects classification accuracy? For instance, could they compare results when preprocessing is applied within the training set only versus the entire dataset? This would strengthen confidence in their conclusions and align with best practices in ML validation.

We thank the reviewer for drawing attention to the topic of data leakage, which we agree is crucial to address. Our intention by commenting on data leakage was not to suggest substantial data leakage especially in our study, but rather to highlight a general gap in the literature: the potential influence of subtle forms of leakage during EEG preprocessing, which have not been systematically examined. We named this particular leakage "latent leakage" in the revised manuscript. Prior studies addressed more common forms of data leakage – such as between-participant leakage (Brookshire et al., 2024) or training-test overlap in feature selection (Kinahan et al., 2024).

In our case, test data were not used for training or feature selection but were involved in global preprocessing steps like autoreject (for threshold estimation) and ICA (for artifact removal). These steps could, in theory, slightly bias the test accuracy by improving artifact correction. However, any such effect would likely lead to higher, not lower, test accuracies. Notably, our results show that applying ICA, autoreject, or low high-pass filters reduced decoding performance, suggesting that the reported influences of those steps would rather be amplified in a leakage-free pipeline.

Fully separating preprocessing per cross-validation (CV) fold would better prevent leakage but is not feasible in our multiverse design: First, trials were not recorded in discrete runs, making leave-one-run-out CV impractical without class imbalance or trial loss. Second, repeating ICA and autoreject per CV fold would increase preprocessing time, as the data needs to be preprocessed 5 times per processing step, one time for each CV fold using the entire train data.

Early epoching could balance folds but risks filter edge artifacts and unstable ICA due to less data.

We see this as an important but currently underexplored topic that warrants a dedicated follow-up study. However, in the revised manuscript, we analyzed leakage (i.e., latent leakage) on an isolated subset of the data. We could not find evidence that this kind of leakage increased test accuracy and any consequently altered interpretations of our results. We have expanded and clarified our discussion of data leakage in the Methods section (page 24, lines 806–848) and included detailed analysis methods and results in the Supplementary Material (Supplementary Text – Influence of latent leakage on decoding performance). As illustrated in Figure S32, we found no evidence of latent leakage.

- Could the authors provide additional details on the architectures and hyperparameters of EEGNet and the time-resolved classifier? Specifically, information on layer structures, activation functions, dropout rates, batch size, learning rate, and training epochs would improve reproducibility. Providing a clear architecture table or a link to code would significantly improve reproducibility.

Following the reviewer's suggestion, we have added more information about model structure and hyperparameters in the Methods section (page 23, lines 780-785; page 24, lines 798-801). Further, we included Supplementary Tables S7 and S8, providing EEGNet hyperparameters and architecture, respectively. Hyperparameters of the time-resolved models were added in Table S9.

- Figure S28, S29 (now S29, S30) demonstrate that autoreject removes different epochs depending on the random seed, introducing stochastic variability in classification results. Could the authors discuss whether results are stable across multiple runs with different seeds? A sensitivity analysis showing whether classification accuracy remains consistent would be valuable.

To avoid repeating the entire multiverse estimation and decoding for several different seeds we sampled some exemplary forking paths from all participants and ran autoreject with different seeds. We computed intraclass correlation coefficients describing the variability of accuracy estimation across different seeds. With respect to the participants, we found excellent intraclass correlations (all ICCs > 0.867), whereas with respect to the forking paths, we found fair intraclass correlations (all ICCs > 0.466, see Supplementary Text – Variability due to random seed in autoreject).

Minor Comments:

- Would the results reported for decoder performance be generalisable to other datasets? Could authors do a sub-analysis to test this, or if this is beyond the scope of the current study, the authors could discuss potential generalisability issues.

We assume that the reviewer is referring to generalizability to data sets where a comparable experiment was used to elicit the respective ERPs. In this case, we do not expect significant changes, even across different electrode montages, online filters, online references, and other variations used during the experiment. Although important, it is beyond the scope of the present study to test this hypothesis across more than the seven data sets already analyzed. As already

addressed in the discussion section (Discussion – Generalizability of the findings from the multiverse), we refrain from generalizing our results to substantially different experimental designs, such as for instance rapid serial visual presentation experiments or resting-state data.

- To improve clarity, could the authors provide a specific example within the citation (21-23) to illustrate the referenced concept?

We apologize for the rather loose reference to "explainable AI" methods, or methods for model interpretability in general. The sentence was specified in the revised manuscript (page 13, lines 370-372).

- To better capture classifier performance beyond accuracy, could the authors include additional metrics such as recall, specificity, AUC, and precision? This would be particularly useful for understanding class imbalances and systematic errors in decoding.

Unfortunately, we did not save other metrics during model estimation. Since the EEGNet estimations are the most computationally intensive step of the entire study, we refrained from repeating them for the entire dataset. We focused on reporting and analyzing balanced accuracy and found it to be an appropriate metric for both the balanced and imbalanced use cases (Thölke et al., 2023).

- To provide a clearer reference point, could the authors include a comparison of decoding performance on unprocessed EEG data versus pre-processed data? This would help quantify the extent to which preprocessing enhances or alters classification outcomes.

In the revised manuscript, we have added the reference forking paths without any preprocessing to the raincloud plots (Fig. 3, Fig. S1), indicated by small triangles, and updated the respective figure captions and text (page 6, lines 149-155).

- Could the authors investigate whether preprocessing choices lead to spurious decoding in baseline periods? This could help determine whether classifiers are relying on residual artifacts rather than meaningful neural signals.

We have now added a formal statistical evaluation of the influence of preprocessing on baseline artifacts. The results indicate, that the use of artifact correction steps reduced the probability of a baseline artifact. We have expanded the Methods (page 28, lines 965-978), Results (page 12, lines 315-326), and Discussion (page 15, lines 430-431) sections and updated Figure S20 with the newly derived p-values.

- Could the authors discuss the trade-offs between maximizing decoding accuracy and maintaining neurophysiological interpretability? This would be particularly useful for ERP researchers who prioritize signal clarity over classification performance.

Following the reviewer's suggestion, we made several changes throughout the manuscript to disentangle decoding performance from interpretability (e.g., page 1, lines 27-30; page 13, lines 359-364 & 373-376; page 15, lines 431-442).

- Could the authors highlight which preprocessing choices are most likely to introduce baseline artifacts (Figure S19, now S20)? This would help researchers avoid overestimating classification

accuracy due to preprocessing-induced distortions.

We have now highlighted the facets in Figure S20 to denote significant effects for each experiment. See above (two comments earlier) for a detailed explanation on how we added formal statistics to the baseline artifact analysis.

Reviewer #2

The paper “How EEG preprocessing shapes decoding performance” explores how different preprocessing steps impact EEG classification using EEGNet and time-resolved logistic regression. By testing multiple preprocessing pipelines across seven ERP experiments, the study shows that choices like filtering, artifact correction, and baseline correction can significantly affect decoding accuracy. Using linear mixed models (LMMs) on T-sum for time-resolved analysis, the authors provide a clear and systematic way to measure these effects. Their approach is rigorous, transparent, and valuable for improving EEG decoding methods, offering practical insights for both research and brain-computer interface applications.

The study provides important insights into EEG preprocessing but is limited by its narrow range of artifact rejection methods, which do not reflect common practices in the field. Many widely used techniques, such as ASR, ICLabel, MARA, and the PREP pipeline, are available only in MATLAB and were not included or replicated in Python, despite MNE being used in only ~5% of EEG studies as of 2025. This omission weakens the generalizability of the findings. Expanding the preprocessing pipeline to include these methods would make the results more representative of real-world EEG research. Other limitations are indicated below.

Limitations and Suggested Improvements

1. Limited Artifact Rejection Methods. The study relies on MNE for artifact rejection, which supports only a small subset of existing methods. Many widely used approaches, such as Clean_rawdata, Cleanline, ASR, ICLabel, MARA, and the PREP pipeline, are primarily implemented in MATLAB and are not available in MNE. Given that MNE is used in only about 5% of EEG publications as of 2025, preprocessing the data in MATLAB or implementing Python equivalents for these methods would have provided a more comprehensive analysis of artifact rejection.

We appreciate the reviewer's comment to extend the analysis to other artifact correction approaches. While we understand the concern regarding the practical relevance of MNE, recent evidence suggests, that its current usage is more widespread than the suggested 5%. For example, in the EEGManyPipelines project (Trübtschek et al., 2024), MNE was the second most common toolbox used by the participating research teams (Cesnaite, 2023) (26%, 44 out of 168), while EEGLAB/ERPLAB taken together were the most prevalent (33%, 55 out of 168). It is also likely that researchers focused on machine learning and decoding are even more inclined toward Python-based toolboxes such as MNE, as compared to MATLAB-based alternatives.

There are several challenges associated with adding a broader range of artifact rejection methods. First, the pipeline currently comes with large computational cost, and simple repeating

the pipeline in an entirely new programming language doubles the processing cost, yet alone the addition of several more different artifact correction methods. Second, adding only the 5 more artifact rejection methods to the current pipeline would increase the number of pipelines by a factor of 5 if the methods are included exclusively (i.e., one of them is used), or by a factor of 25 if they are all added in a on/off fashion. These estimates are conservative, assuming only a single variant of each method is tested, whereas our current implementation of autoreject, for example, includes two sets of hyperparameters ("reject" and "interpolate"). Next, when applied in an exclusive fashion, all competing artifact rejection methods should be placed on the same hierarchical level in the multiverse to not break the intuition or interpretations of the multiverse analysis. However, we kindly ask for the reviewer's understanding that a comprehensive evaluation of the wide variety of such methods is beyond the scope of the current study and is more appropriately undertaken in future research using a systematic on/off comparison against well-defined reference pipelines, such as done for instance in Delorme (2023). We have added the need for future research with an alternative design on an extended set of artifact correction toolboxes in the discussion section (page 18, lines 568-573).

2. LMM Significance and Figure 5 Improvements. Linear Mixed Models (LMMs) inherently control for multiple comparisons, so additional corrections are unnecessary.

Following the reviewer's suggestion, we have removed the correction for multiple comparisons in the F-tests. We have deleted the part in the Methods (page 27, lines 949-953) and updated Supplementary Tables S1 and S2. Further, we used the updated p-values to highlight significance in the heatmaps and interaction plots (see below).

Figure 5 should highlight only significant differences rather than showing all variations in color. Additionally, a row grouping all experiments together would help visualize overall trends.

We have updated the plots by coloring only the steps that became significant (Fig. 5 and S2). We have decided against adding an extra column in the updated figure. An extra column would indicate that some statistics were performed across models (see next comment), especially now that panels are only colored if a step is significant. We believe that a trend across experiments can be visually inferred from the rows.

The rationale behind integrating all experiments into a single LMM model needs clearer explanation in the methods section.

We recognize that our explanation of the model construction may have been unclear and would like to clarify this point. We fitted one separate linear mixed model or linear model per experiment. This approach offered several advantages. First, it reduced the number of parameters and data points, making model estimation more tractable. Second, separate models avoid the need to include complex interaction terms with the experiment factor, which is important given that many effects—and their magnitudes—are specific to individual experiments. To improve clarity, we have revised the relevant sections of the manuscript (page 8, lines 174-179; page 26, line 899 & 914).

3. ICA Application and Artifact Detection. The application of Independent Component Analysis (ICA) in the default pipeline is suboptimal. Standard EEG preprocessing applies ICA after filtering, which was only tested in an alternate multiverse but should likely be the default.

We agree with the reviewer that at least a high-pass filter (HPF) should be applied before ICA. In fact, this is done in our framework with a 1 Hz HPF (Winkler et al., 2015) by applying the HPF to a copy of the data, on which the components are then estimated and classified. The selected components are then removed from the raw, unfiltered time series, which is then further preprocessed in the multiverse. This procedure is in accordance with the tutorial in the MNE documentation¹. The procedure applied in the current study also allows us to reduce the number of "nonsensical" preprocessing paths in the multiverse context, i.e., by avoiding that the ICA solution relies on the choice of the HPF of a forking path. As the reviewer pointed out, computing ICA on unfiltered data could lead to an inadequate ICA solution, affecting all forking paths without filtering but with ocular ICA. By applying the MNE tutorial-guided approach, as done in the present study, we decouple the ICA approach to make its influence independent of the HPF chosen in the respective forking path.

In the "alternative multiverse" to which the reviewer refers, we applied the ICAs after the forking path-specific filtering. There, we still followed the procedure of filtering at 1 Hz on a copy of the data. However, the destructive effects of the filter on the muscle ICA are profound, and we therefore decided to use the multiverse approach reported in the main manuscript with the ICA decoupled from the multiverse-filtering. We discuss the consequences of the muscle artifact correction after the forking path-dependent filtering in the supplementary material and specified the use of the 1 Hz filter there as well (Supplementary text – Alternative order of preprocessing steps, and corresponding Fig. S27).

Furthermore, artifact detection based on thresholding activity is simplistic and not used in practice—modern EEG pipelines use machine learning-based ICA selection methods like ICLabel. To reflect this limitation, the study should rename ICA as “Naive ICA” to clarify that it does not represent the current state-of-the-art approach. This point should also be mentioned in the discussion.

We agree with the reviewer that there are more sophisticated ICA-based approaches to selecting the artifactual components, but we deliberately kept our approaches close to the standard implementation of the corresponding toolbox. Changing the ICA version to a more sophisticated one (with presumably higher accuracy in artifact detection) would likely not change our interpretations: Removing the artifacts that are systematically associated with the decoded category could reduce the decoding accuracy even better if the artifacts are better detected, thus strengthening the reported main effects. However, we believe that characterizing the method as 'naive' may overlook its widespread adoption and practical utility in current and past EEG preprocessing workflows. We have added a distinction of the currently applied, "threshold-based" ICA to other ICA-based methods in the Methods (page 20, lines 658-659, page 21, lines 678-680) and Discussion sections (page 13, lines 374-376, page 17, lines 547-557).

4. Potential Overfitting in EEGNet Training. EEGNet is trained with a fixed 200 epochs, but no evidence is provided to confirm that this choice prevents overfitting. Some validation data should be presented to justify this parameter.

In the multiverse setting, we used only training and test sets, without separate validation sets. Validation sets would be necessary especially for hyperparameter optimization, which was not

¹ https://mne.tools/1.5/auto_tutorials/preprocessing/40_artifact_correction_ica.html

computationally feasible for the high number of forking paths and participants. However, we decided to use some default hyperparameters and selected a "batch size" of 16 and "max epochs" of 200 based on piloting in previous projects. Our EEGNet models are likely to overfit to train data, but we only saved and report test accuracy (averaged across CV splits) to quantify the ability to generalize to unseen data. If we were to perform hyperparameter optimization, we would expect to be able to address overfitting on the train data, although there is still a low number of trials compared to a high number of estimable model parameters (>1000). However, hyperparameter optimization would likely result in different optimal hyperparameters per forking path, highly aggravating the interpretation on multiverse results. A similar intuition applies to time-resolved decoding. We did not perform hyperparameter optimization and report only test accuracies. Since one model is fitted per time point, it would be even more complex to investigate overfitting.

Additionally, other EEGNet training parameters are not disclosed, making it difficult to replicate the results.

We thank the reviewer for catching this. In the revised manuscript, we have added information about model structure and hyperparameters (page 23, lines 780-785; page 24, lines 798-801). Further, we added Supplementary Tables S7 showing EEGNet hyperparameters, Tables S8 showing EEGNet architecture, and Table S9 showing the hyperparameters of the time-resolved models.

The version of BrainDecode used should also be specified.

We have clarified the version numbers of braindecode and other key programming languages and toolboxes in the different parts of the Methods section. We also reviewed and updated these entries where needed. Additionally, a complete list of package versions (including Python, R, and Julia) is now provided in the GitHub repository. The code availability statement has been updated to include references to those lists.

5. In the abstract, it is stated, "All artifact correction steps reduced decoding performance across all experiments and models, while higher high-pass filter cutoffs." This conclusion aligns with that of reference 33, "Delorme, A. EEG is better left alone. *Sci. Rep.* 13, 2372 (2023)." It is important to emphasize this point in the discussion, particularly given that this paper incorporated a broader range of artifact rejection methods and EEG references, suggesting that the current approach might generalize to other rejection methods.

We thank the reviewer for referring to this study. While we find converging evidence regarding the influence of filtering, the findings related to artifact correction appear more mixed. Although their abstract suggests that automated artifact rejection negatively impacts the number of significant channels, the Results section indicates a more nuanced picture: some methods increased the number of significant channels, others had no effect, and others decreased the number of significant channels. For some artifact correction approaches, the outcomes varied across experiments – possibly due to more heterogeneous data sources or smaller samples – making it difficult to draw detailed comparisons. We extended the discussion with their study in our revised manuscript (page 15, lines 443-454; page 16, lines 503-504).

6. Software Reproducibility and Filtering Details. The study does not document the versions of MNE, Autoreject, or other key software used, which is critical for reproducibility.

We provide and updated software and toolbox version information for the most essential tools in the Methods section (i.e., Python: MNE, autoreject, Braindecode, scikit-learn; Julia: MixedModels; R: rCall, JellyMe4, afex, emmeans). A comprehensive list of all package versions is now available in the GitHub repository, and we have added a direct link in the Code Availability Statement for clarity and ease of access.

Additionally, filter characteristics (e.g., filter order, roll-off, and type) should be explicitly stated to ensure precise replication.

We thank the reviewer for catching this. In response, we have now provided more detailed specifications of the filters in both the Methods section (page 21, lines 684-690; page 20, line 660; page 21, lines 678-680) and the in Table S6.

7. Cross-Validation and Confidence Intervals. The study uses 5-fold cross-validation, but confidence intervals are not provided in Figure 6. Adding confidence intervals would clarify the variability in classification accuracy and improve result interpretation.

We thank the reviewer for highlighting the importance of variability in decoding performances. During the estimation of decoding accuracy, 5- or 10-fold cross-validation was applied for EEGNet and time-resolved classifiers, respectively, with results averaged accordingly. However, the single accuracy values per fold would not be statistically informative. For the averaged decoding results, we employed classical statistical methods by estimating marginal means within linear and linear mixed models. Unfortunately, the inherent confidence intervals in the context of the R package *emmeans* are misleading and not recommended², and they also increase the visual complexity of the plot. To address this issue, we have added asterisks indicating the significance of the *F*-test in each facet (and plot legends). This affects Figure 6 and Figures S6-S18. We have also added an explanation to the respective figure captions.

8. Alternative to T-sum: Threshold-Free Cluster Enhancement (TFCE). Instead of using T-sum for statistical testing in time-resolved decoding, the study could adopt Threshold-Free Cluster Enhancement (TFCE). TFCE improves sensitivity and avoids arbitrary cluster thresholds, making it a more robust alternative to traditional cluster-based permutation testing (CBPT).

We are grateful for the reference to the TFCE method, which can eliminate the dependence on the cluster-forming threshold. To our understanding, the method still requires the CBPT on the retrieved TFCE time series. Following the reviewer's suggestion, we repeated the group-level statistics using TFCE and calculated cluster p-values using permutation mass tests. To find a summary value across time points, we summed the TFCE values for all significant clusters, similar to what was done previously with T-sums. Since we found high correlations for all experiments (all $r > 0.96$) between T-sum and TFCE-sum across forking paths, similar results regarding the effect of preprocessing steps on decoding performance for the two methods (fixed cluster-forming threshold vs. TFCE) can be expected. We have added this analysis to the revised manuscript (page 26, lines 890-896) and added Figure S33.

9. Availability of Raw EEGNet and Time-Resolved Decoding Results for Further Analysis: The study presents a rich dataset with extensive decoding results from both EEGNet and time-

² <https://cran.r-project.org/web/packages/emmeans/vignettes/comparisons.html>

resolved logistic regression across a large multiverse of preprocessing pipelines. However, the raw classification results are not available. Releasing these raw results would provide a valuable resource for the community.

The Data and Materials Availability statement has been updated to direct readers to the Zenodo repository, where the large files containing the individual classification results are available.

Reviewer #3

This article reports the results of a fantastically detailed, well thought out, and extensive study of the effects of a good number of EEG preprocessing steps on the decoding of event-related potentials from cognitive task-related EEG. The methods used are robust, as are the statistical tests. Conclusions drawn are accurate, although I have suggested a few additional caveats and considerations. I have also suggested a number of considerations for the authors below. These considerations do not indicate that the study is not already excellent, they are merely suggestions for improvements, provided from my perspective as a researcher experienced with EEG preprocessing and ERP analysis.

Abstract:

- Suggest stating that other pre-processing steps were optimal for specific experiments / event-related potential components (because it's not possible to discern between whether it's the influence of preprocessing on the experiment or on the specific ERP).

We have rephrased the sentence as suggested by the reviewer (page 1, lines 24-25).

- While it is true that all artifact correction steps reduced decoding performance across all experiments and models, I would suggest that a strong caveat be provided that from the current results, it's impossible to tell if this might be the influence of artifacts in the data. Although in BCI applications it may be useful, if decoding studies are implemented to tell us more about neural activity, then higher decoding accuracy in the context of higher artifact influence is counter-productive rather than useful. Sole focus on decoding accuracy is an issue here, in a similar way to studies that emphasize high classification accuracy at the expense of data leakage (and thus less practically useful results). I suggest statements to these effects be provided strongly and clearly in the abstract, so future research does not emphasize high decoding accuracy at the expense of practical usefulness as a result of the influence of this manuscript (the statement "if not corrected, artifacts facilitate decoding but compromise conclusive interpretation" is not strong or clear enough).

We agree that relying on decoding performance alone is not enough to choose the appropriate data analysis pipeline. Additional emphasis has been placed on this point throughout various sections of the revised manuscript. Further, we rephrased the sentence in the abstract according to the reviewer's suggestion (page 1, lines 27-30).

Introduction:

- Line 38 – similar to my comments on the introduction, I would caveat these statements with the note that maximizing decoding performance is not useful, even in contexts with

compromised data quality, if the maximising comes at the expense of validity or the potential to lead to practical utility. E.g. it would be trivially easy to maximise decoding performance by asking participants to blink on certain trials and not others, but not practically useful.

We fully agree with the reviewer and have rephrased and expanded the statement (page 1, lines 45-49).

- I'd suggest expanding a bit on how decoding can be useful (for more than just BCI and reducing the multidimensional space of EEG data).

We have now included additional points while keeping the paragraph concise (page 1, lines 35-42).

- Line 63 – suggest briefly explaining how each of the EEGNet and time-resolved logistic regression classifier works for readers who might be less familiar with them. Particularly for the logistic regression classifier, since the reference provided for this doesn't seem to describe the method (Gramfort, A. et al. MNE software for processing MEG and EEG data. *NeuroImage* 86, 446–460 (2014)).

We have now added one explanatory sentence per decoding framework in the Introduction (page 2, lines 69-75). Following the reviewer's observation regarding the unspecific reference, we have also added a more specific source for time-resolved decoding (King et al., 2018), while maintaining the reference to the MNE package in which the applied version is implemented.

Results:

- Line 162: I think it's worth noting in the results that it isn't clear whether removing EOG (and muscle) artifacts decreases decoding performance because: 1) the artifacts are informative of differences between conditions, or 2) removing the artifacts also removes informative neural activity. Recent research has suggested that removing EOG artifacts with ICA does also remove neural activity due to the imperfect separation of neural and artifact source activity by ICA: Bailey, N. W., Hill, A. T., Godfrey, K., Perera, M. P. N., Rogasch, N. C., Fitzgibbon, B. M., & Fitzgerald, P. B. (2024). EEG is better when cleaning effectively targets artifacts. *bioRxiv*, 2024-06. While this might typically be considered a discussion point, I suspect some readers may not read the entire article, and this point is fairly critical for consideration in future research, so I think it is worth the authors taking steps to ensure it is not missed.

We thank the reviewer for referring to this study. For better readability, we have added the reference and comments at several points in our revised Discussion section, also according to the reviewer's later comments (page 13, lines 359-364; lines 373-376).

- Line 233: Suggest re-writing the following sentence for readability: "Several effects can be observed when using autoreject in the version, in which noise-contaminated trials were discarded rather than interpolated. When not applying a HPF, linear detrending or baseline correction but using autoreject in this version, decoding accuracies were lower"

We have rephrased the sentence as suggested by the reviewer (page 10, lines 264-266).

- Figure 6 – I think the y-axis should be labelled as "Balanced Accuracy" rather than marginal mean?

We thank the reviewer for pointing this out. We have relabeled the figure captions and y-axes as "Accuracy" (all EEGNet interaction plots, Fig. 6 and Supplementary Figures) or "T-sum" (all time-resolved interaction plots and Supplementary Figures). For consistency, we used "Accuracy" in the figure axes, but described it in the text as the "averaged, cross-validated, (balanced) test accuracy". We also changed the description in the captions of Figure 6 and S6 and added "(Balanced)" to the caption of Figure 4.

- Line 256 – As far as I can tell, the authors haven't specified whether causal or acausal filters were applied. If acausal filters were applied, this could explain some of the baseline period decoding accuracy, with the acausal filter smearing weights to influence activity at earlier timepoints (in contrast to the causal filters where filter weights only influence subsequent timepoints). This issue should be explained in detail, as it likely affects many decoding studies.

We thank the reviewer for their comment on the filtering. We have now added a more detailed description in the Methods section (page 21, lines 684-690; page 20, line 660; page 21, lines 678-680), and Table S6 illustrating the filter properties for the different forking paths. This is indeed an important point for discussion. As in most of the preprocessing steps, we used the "default" implementation of the MNE toolbox, closely following the respective tutorial³. By using these non-causal filters, we allowed the filter to smear the signal in both directions instead of just one, potentially affecting the baseline as well. We extended explanations in Results (page 12, lines 324-326) and Discussion (page 14, lines 399-402).

- Line 276 – should it be 0.4 seconds rather than 0.4ms pre-response? (0.4ms would be practically the same as just using the response timepoint).

It should indeed be 0.4 seconds. We have corrected the typo in the manuscript. In fact, we moved the sentence to the Methods section because the methods for analyzing baseline artifacts were expanded during the revision (page 28, line 970).

Discussion:

- Line 312 – it is true that artifact correction steps impaired decoding. But stating the result like this, without noting that remaining artifacts in the data reduces the practical utility of the decoding analysis, emphasizes decoding accuracy over usefulness. This is common across machine learning applications, but should be strongly resisted. High accuracy is only worth reporting if it is useful, and lower accuracy that is more informative/useful should be preferred over high accuracy that is only the result of artifact contamination.

The situation here is complicated by the lack of ability to determine whether it is the artifact itself driving the increased accuracy, or protection against reducing the neural activity as well as the artifact in the artifact cleaning approach. The fact that both the ICA and the autoreject steps lead to lower decoding accuracy may suggest the artifact itself is inflating accuracies, but without certainty. If the authors are able, it would be good to explore the potential explanations further. If not, they might recommend future research explore the point. Future research might use methods that specifically target eye and muscle artifact activity within the ICA decomposition to enable this (which better protects the neural activity outside of those artifacts), for example the methods explained in: Bailey, N. W., Hill, A. T., Godfrey, K., Perera, M. P. N., Rogasch, N. C.,

³ https://mne.tools/1.6/auto_tutorials/preprocessing/30_filtering_resampling.html

Fitzgibbon, B. M., & Fitzgerald, P. B. (2024). EEG is better when cleaning effectively targets artifacts. bioRxiv, 2024-06.

We fully agree with the statement that decoding performance alone is not a sufficient criterion for many use cases to decide on a preprocessing pipeline. We have taken care to emphasize this point throughout the manuscript and have added further comments based on the reviewer's suggestions. However, we believe that it is still a useful metric to illustrate the impact of preprocessing through the various analyses performed throughout the study. Investigating the unique contribution of signal and artifact in decoding is beyond the current scope, but will be investigated in the future with a different study design and data specifically tailored for this question. We have added a note for further research on this question and also referred to the reference provided by the reviewer (page 13, lines 359-364; lines 373-376).

- On this point, perhaps the best combination of preprocessing steps for decoding analyses that are aimed at understanding / decoding neural activity are ranked ~100 to ~300 in Figure S18 (now S19) – the steps that provide high decoding accuracy, but don't show decoding accuracy where there shouldn't be any (prior to the stimulus presentation, where neural activity cannot be responding to the different conditions). If I were writing the paper, I would emphasize this point.

We agree that selecting these forking paths would minimize the risk of baseline artifacts; however, they might still contain other artifacts which did not smear into the baseline period. While excluding all forking paths with baseline artifacts may effectively remove several unwanted processing combinations, we believe that relying solely on this step may not be enough to ensure that the signals are free of artifacts.

- Line 316 – I'm glad that the authors have addressed this point. However, as noted in my earlier comments, I think the point needs to be highlighted throughout the manuscript. As it currently reads, the rest of the manuscript seems to emphasize decoding performance over the insights into neural activity that might be offered by decoding.

We recognize that our wording may have unintentionally placed more emphasis on decoding performance rather than neural interpretability, in part due to the outcome metric selected as a benchmark. To avoid evaluative wording, we replaced words like "optimal" or put them more in context. Similarly, we replaced wordings like "best performance" with "highest performance", "performed weakly" with "showed lower decoding performance", "best forking path" with "forking path with highest decoding performance", "worst decoding performance" with "lowest decoding performance", "well-performing" with "higher-ranking", "impaired decoding" with "reduced decoding performance" and similar throughout the manuscript. Further, we added another paragraph in the Discussion about the limitations of a multiverse analysis bound to an outcome metric (page 15, lines 431-442).

- Line 321 – Bailey et al., (2024) could be cited to support the point that neural signals of interest are also removed by removing artifacts.

We have added the reference at the corresponding location.

- Line 481 – I think the point about artifacts producing higher decoding performance, but this not necessarily reflecting neural activity needs to be noted here also.

According to the reviewer's suggestion, we have expanded the paragraph (page 18, lines 578-581).

Methods:

- Line 525: Does subtracting the EOG channels from each other adversely affect the ICA performance? From my understanding, this is not a typical approach, and using the same reference for all electrodes included in the ICA is important. If these EOG channels were not included in the ICA (which seems to be the case when the ocular artifact correction is explained later), then this should be explicitly stated.

The two channels that were subtracted exhibit opposite polarity⁴ for eye movements. The rationale for subtraction is to increase the amplitude of the respective EOG relative to other signal sources, such as neural signals and noise common to both EOG channels. This has been done in previous studies (Barbara et al., 2023; Maier et al., 2025), and is implemented in some EEG preprocessing toolboxes (e.g., hu-neuro-pipeline⁵). We excluded the virtual EEG channels from the ICA estimation. The bipolar EOG signals were only used for correlation with the independent components to determine which components reflect eye artifacts. We have added the references to the revised manuscript and specified, that only EEG channels were used for ICA (page 20, line 661).

- Line 560: The work introducing the picard ICA method should be cited.

We have added the reference to Ablin et al., 2018.

- Line 557: While high pass filtering data at 1Hz prior to the ICA improves ICA decomposition performance, and the approach is somewhat common in the literature, high pass filtering data prior to the ICA, then applying the ICA weights back to the unfiltered data has been shown to lead to worse blink artifact correction performance, and weaker ERP effect sizes. This is likely because there are EOG artifact signals <1Hz, which are not addressed by applying the 1Hz filtered ICA to the unfiltered data. Evidence of this is provided in the supplementary materials of: Bailey, N. W., Hill, A. T., Biabani, M., Murphy, O. W., Rogasch, N. C., McQueen, B., ... & Fitzgerald, P. B. (2023). RELAX part 2: A fully automated EEG data cleaning algorithm that is applicable to Event-Related-Potentials. *Clinical Neurophysiology*, 149, 202-222. I don't expect the authors to re-analyse all of their data based on this point, but it is worth noting as a limitation, with the suggestion that the decoding performance after ICA to correct EOG may have been reduced because of this issue.

We thank the reviewer for bringing this study to our attention. We have added the points to our manuscript (page 20, lines 666-670; page 17, lines 555-557).

- Line 601: Subtraction baseline corrections are likely not optimal for ERPs, as they transpose topographical effects from the baseline period into the active period, and do not actually provide the control for baseline shifts that they are intended to achieve. See the following study for details: Alday PM. How much baseline correction do we need in ERP research? *Extended GLM model can replace baseline correction while lifting its limits. Psychophysiology*. 2019;56(12):e13451. Also see the following study for evidence that this can produce both false

⁴ e.g., http://www.eegpedia.org/images/3/3f/Lateral_eye_movement_%28source%29_eegpedia.png

⁵ <https://github.com/alexenge/hu-neuro-pipeline>

positives and false negatives in the ERP active window of interest in real data: Hill, A. T., Chung, S. W., Emonson, M., Corcoran, A. W., Fitzgibbon, B. M., Fitzgerald, P. B., & Bailey, N. W. (2024). Neural differences in conflict monitoring and stimulus expectancy processes in experienced meditators are likely driven by enhanced attention. *bioRxiv*, 2024-12. Similar to the issue with 1Hz high pass filtering prior to ICA, I do not expect the authors to re-analyse their data based on this issue, but it should be noted, with suggestion that future research explore regression baseline correction methods with decoding analyses to see if outcomes are improved over subtraction baseline correction methods.

We agree that this would be interesting to test also using decoding. We added a comment and the references to the Discussion (page 17, lines 544-546).

- Line 676: With the statement: "Data of each time point were standardized by removing the mean and scaling to unit variance." – please indicate whether the scaling is applied per timepoint, across timepoints, including just each individual separately, or all individuals, etc.

In the revised manuscript, we have specified the standardization procedure in more detail (page 23, lines 794-796).

- Line 708: I mostly agree with the sentiment in the point about data leakage. However, it will certainly be worth noting that some preprocessing conditions include more potential data leakage than others, so if data leakage has affected decoding performance, then the comparisons between those conditions and conditions that contain fewer steps that might introduce data leakage might be biased.

We agree that there would indeed be a bias when comparing the respective steps with and without leakage. In our revised manuscript, we have investigated the type of leakage, to which we referred to in our original manuscript, more thoroughly. We tested whether this "latent leakage" actually leads to higher accuracies by isolating and "sealing" certain processing steps (HPF, ocular ICA, autoreject) in a subset of forking paths (Supplementary Text – Influence of latent leakage on decoding performance). We did not find evidence for higher decoding accuracies due to latent leakage in our study (Fig. S32). We have expanded our comment about leakage in the method section (page 24, lines 806-848) including the potentially bias of comparisons when using "leaky" versus "sealed" forking paths (page 24, lines 823-825).

Supplementary Materials:

- Figure S3 – should there be numbers in this figure like in the preceding figure? I'd suggest explaining why the boxes are greyed out also (I assume because they reflect matches to the reference pipeline?).

We thank the reviewer for bringing this to our attention. In the revised manuscript, numbers were added as in the previous figures. As the reviewer correctly assumed, the boxes were greyed out because they match the reference forking path. We added an explanation to the figure caption.

- Figure S4 – great figure, but I'd suggest expanding the y axis so that A and B separately take up the whole page, so that readers can more easily see which steps are included in the top ranked pipelines.

We have split the figure into two separate figures and changed the reference to those in the text accordingly (Fig. S4 & S5).

- Figure S5 (now S6) – it's surprising that simply subtracting Cz from all the signals inflates decoding performance so much compared to average or P9/10 references. This suggests that the Cz signal itself is highly informative. Perhaps this should be noted in the discussion as a point for further exploration in future research.

We thank the reviewer for this observation. This is also consistent with what is shown in Figure 5. It is also consistent with Kappenman et al., (2021 & their Fig. S2). Cz is also close to the electrode of interest in ERN studies such as FCz (Kappenman et al., 2021), suggesting a similar response profile. We have added a comment on this in the Discussion to further acknowledge the limitations of interpretability that may be associated with the use of this electrode (page 16, lines 468-474).

- Figure S6 (now S7) onwards – I think the authors should replace “see figure 6 for details” with an explanation suited to each figure, to make it easier for the reader to understand each figure in its own context (rather than having to refer between the main manuscript and the supplementary materials to understand each supplementary materials figure). Also, I think the y-axis are mislabelled for at least some of these figures.

The axes were indeed mislabeled. We changed the labels to "Accuracy" and "T-sum" instead of "Marginal mean" in all plots showing interactions. Further, the caption of Figure 6 was updated and Figures S6 and S12 were given self-explanatory captions without the need to jump between manuscript and Supplementary Information. To avoid redundancy, subsequent figures captions refer to Figures S6 (EEGNet) and S12 (time-resolved) for more details.

- Figure S19 (now S20) - Even if very few forking paths produced baseline artifacts, if the goal is to decode conditions based on neural activity, any baseline artifact at all suggests that goal is not being achieved, and performance is suspect to contamination by non-neural sources. As such, even for the later ERPs (N400 and P3) which should show low baseline decoding artifact, removing artifacts with ICA seems like a critical step. The issue of decoding performance produced by non-neural sources should be considered more critical than low decoder performance. I view this as analogous to data leakage – high performance doesn't matter if performance is due to an artifactual signal, regardless of whether the artifact is in the classification algorithm design or the signal processing pipeline.

We agree with the reviewer, that artifact correction steps are helpful to reduce baseline artifacts, and have added a more detailed analysis of preprocessing steps that facilitate baseline artifacts. We have expanded the Methods (page 28, lines 965-978), Results (page 12, lines 315-326), and Discussion (page 15, lines 430-431) sections and updated Figure S20 with the newly derived p-values.

- Figure S25 (now S26) – Based on the description of this figure early in the supplementary materials, I think it would be worth noting in the results and discussion the point that a lower low-pass filter might replace muscle artifact rejection to enable higher decoding performance (which also eliminates the concern that the muscle artifact itself is leading to higher decoding performance).

Based on the reviewer's comment, we have added a note to the Discussion (page 14, lines 392-393).

- Figure S25 (now S26) – The figure caption should explain what this alternative multiverse reflects (the different ordering of preprocessing steps?)

We have added an explanation to Figures S26 and also S27.

- Figure S29 (now S30) – is this an example from just one participant? I think the caption could explain more clearly.

The assumption that it was based on an example participant was correct, but we only included the explanation it in the Supplementary Text. We have expanded the captions of Figures S29 and S30 accordingly.

- Figure S31 (now S34) – the title of one of the panels in the top part of the figure seems to be wrong (given the values in the two panels are reverse polarity from each other).

We are confident that the panel labels are accurate. AIC values can be either positive or negative, and comparisons are only meaningful between models using the same dataset, i.e., between adjacent gray and black bars (or the connected gray and black dots in the updated figure). Lower (or more negative) AIC values indicate better models. However, we acknowledge that the bar plots may be misleading, particularly due to the unnecessary inclusion of zero as a reference point, which is not relevant in this context. In response, we have revised the figure type of Figure S34 and updated the caption accordingly.

- Table S1 – typo – “significance”.

We have corrected the typo.

Further changes (reviewer-independent):

- We have added an additional Code availability statement and removed the part from the Data and materials availability statement, according to the editorial policy checklist.
- Acknowledgments, Author contributions, Competing interests, Data and materials availability statements and Code availability statements were moved before the References, according to the editorial policy checklist.
- We have lowered the dot-size in the correlation plots of random intercepts for aesthetic reasons (S24, S25).
- Descriptions about the calculations of the boxplots have been added to Figures 3, S21, S22, S26.
- We have added a space (old: "p= x.x", new: "p = x.x") to the p-values in the plot annotations and increased plot width of S22 and S23.
- We included a reference to a previous study showing that there is no general participant-specific cross-experiment correlation of decoding accuracies (page 13, lines 344-346).

References

- Ablin, P., Cardoso, J.-F., & Gramfort, A. (2018). Faster Independent Component Analysis by Preconditioning With Hessian Approximations. *IEEE Transactions on Signal Processing*, *66*(15), 4040–4049. <https://doi.org/10.1109/TSP.2018.2844203>
- Barbara, N., Camilleri, T. A., & Camilleri, K. P. (2023). Monopolar and bipolar electrooculography signal characteristics due to target displacements—Have we seen the whole picture? *Physiological Measurement*, *44*(3), 035011.
- Brookshire, G., Kasper, J., Blauch, N. M., Wu, Y. C., Glatt, R., Merrill, D. A., Gerrol, S., Yoder, K. J., Quirk, C., & Lucero, C. (2024). Data leakage in deep learning studies of translational EEG. *Frontiers in Neuroscience*, *18*, 1373515. <https://doi.org/10.3389/fnins.2024.1373515>
- Cesnaite, E. (2023, October). *EEGManyPipelines: Robustness of EEG results across analysis pipelines. Cutting Gardens Conference* [Conference Presentation]. Cutting Gardens Conference, Frankfurt. <https://osf.io/r62na>
- Delorme, A. (2023). EEG is better left alone. *Scientific Reports*, *13*(1), 2372. <https://doi.org/10.1038/s41598-023-27528-0>
- Kappenman, E. S., Farrens, J. L., Zhang, W., Stewart, A. X., & Luck, S. J. (2021). ERP CORE: An open resource for human event-related potential research. *NeuroImage*, *225*, 117465. <https://doi.org/10.1016/j.neuroimage.2020.117465>
- Kinahan, S., Saidi, P., Daliri, A., Liss, J., & Berisha, V. (2024). Achieving Reproducibility in EEG-Based Machine Learning. *The 2024 ACM Conference on Fairness, Accountability, and Transparency*, 1464–1474. <https://doi.org/10.1145/3630106.3658983>
- King, J.-R., Gwilliams, L., Holdgraf, C., Sassenhagen, J., Barachant, A., Engemann, D., Larson, E., & Gramfort, A. (2018). *Encoding and Decoding Neuronal Dynamics: Methodological Framework to Uncover the Algorithms of Cognition*. <https://hal.science/hal-01848442>
- Maier, M., Leonhardt, A., Blume, F., Bideau, P., Hellwich, O., & Rahman, R. A. (2025). Neural dynamics of mental state attribution to social robot faces. *Social Cognitive and Affective Neuroscience*, nsaf027.
- Thölke, P., Mantilla-Ramos, Y.-J., Abdelhedi, H., Maschke, C., Dehgan, A., Harel, Y., Kemtur, A., Mekki Berrada, L., Sahraoui, M., Young, T., Bellemare Pépin, A., El Khantour, C., Landry, M., Pascarella, A., Hadid, V., Combrisson, E., O'Byrne, J., & Jerbi, K. (2023). Class imbalance should not throw you off balance: Choosing the right classifiers and performance metrics for brain decoding with imbalanced data. *NeuroImage*, *277*, 120253. <https://doi.org/10.1016/j.neuroimage.2023.120253>
- Trübtschek, D., Yang, Y.-F., Gianelli, C., Cesnaite, E., Fischer, N. L., Vinding, M. C., Marshall, T. R., Algermissen, J., Pascarella, A., Puoliväli, T., Vitale, A., Busch, N. A., & Nilsson, G. (2024). EEGManyPipelines: A Large-scale, Grassroots Multi-analyst Study of Electroencephalography Analysis Practices in the Wild. *Journal of Cognitive Neuroscience*, *36*(2), 217–224. https://doi.org/10.1162/jocn_a_02087
- Winkler, I., Debener, S., Müller, K.-R., & Tangermann, M. (2015). On the influence of high-pass filtering on ICA-based artifact reduction in EEG-ERP. *2015 37th Annual International Conference of the IEEE Engineering in Medicine and Biology Society (EMBC)*, 4101–4105. <https://doi.org/10.1109/EMBC.2015.7319296>

Response to reviewers

We again thank all three reviewers for their thoughtful feedback on our revised version of the manuscript. Below, we present a response to the remaining discussion points, with our replies highlighted in blue. Our provided line numbers correspond to the manuscript version with track changes.

Reviewer #2

Regarding my earlier comment on the Limited Artifact Rejection Methods, I do not believe it was adequately addressed in the rebuttal. While I understand the computational constraints involved in expanding the pipeline to include more artifact rejection methods, this is not a sufficient justification for omitting the most widely used techniques in the field—such as `clean_rawdata`, `ICLabel`, `MARA`, and `PREP`. These methods are integral to current EEG preprocessing practice, and excluding them limits the generalizability of the findings.

Given this methodological restriction, the current title “How EEG preprocessing shapes decoding performance” overstates the scope of the study. To more accurately reflect the specific implementation context, I strongly recommend revising the title to something similar to “How MNE-based EEG preprocessing shapes decoding performance.” This change would provide appropriate transparency about the methodological boundaries and prevent misleading readers regarding the study’s generalizability.

That said, I appreciate that my other comments were addressed thoroughly and constructively in the revised manuscript.

We thank the reviewer for their thoughtful follow-up and for recognizing the revisions made in response to the earlier comments. We appreciate the concern regarding the breadth of artifact rejection methods and the potential implications for generalizability.

That being said, we believe that many of the preprocessing steps included—such as high- and low-pass filtering, re-referencing, or baseline correction—have direct counterparts in other toolboxes (e.g., `EEGLAB` or `FieldTrip`). For these standard steps, we would not expect substantial differences in the outcome of our multiverse analysis purely based on the software implementation. Nonetheless, we agree that conceptually different approaches to artifact rejection, like those used in the toolboxes mentioned by the reviewer, may lead to different results in the multiverse context. If the toolboxes however merely correct the artifacts more thoroughly, we expect our interpretations to be similar, as discussed already in our last revision (e.g., page 17, lines 538-547).

To better reflect the study’s methodological boundaries, we have revised the abstract to clearly highlight that all analyses were conducted using `MNE-Python`. While we believe the main conclusions are relevant beyond the specific implementation, this clarification ensures transparency for readers and allows them to interpret the findings within the appropriate context.

We appreciate the reviewer's suggestion to revisit the title. However, we feel that the current title accurately reflects the main scientific question, and the change in the abstract should clarify the scope sufficiently.

Once again, we thank the reviewer for their constructive input, which has helped us improve the clarity and framing of our manuscript.

Reviewer #3

Thank you for addressing my previous comments, I think the manuscript is improved by the changes, and am satisfied that it now accurately presents the results and conclusions from this very interesting study.

Minor points remaining:

Page 17 and 20: the explanation on page 20 that copying the 1Hz high pass filtered ICA decomposition to the raw data or data with a lower high pass filter setting may lead to poorer artifact reduction is not explained very clearly, and I cannot see where the adjustment has been made on page 17 to address this point.

We thank the reviewer for reiterating this concern. Our explanation on page 20, lines 655 ff. (in the updated manuscript with track changes) was not very detailed. In accordance with the results from the article mentioned previously by the reviewer¹, we agree that the widely used technique of applying ICA to a 1 Hz high-pass filtered version of the data can lead to worse ocular artifact correction by missing the ocular signal in the <1 Hz frequency range. While this can decrease the effect size of ERPs, the direct effect on decoding accuracy in our study is unclear. On the one hand, the residual artifact could simply represent "noise" obscuring the predictive signals and weakening decoding performance, as the reviewer stated. On the other hand, in agreement with the results illustrated in Figure 5 of our manuscript, the remaining artifacts could lead to an overestimation of decoding accuracy if they were systematically associated with the classified experimental condition. We have revised the sentences accordingly on page 20, lines 655 ff.

We apologize for the imprecise reference to page 17, lines 555 ff. The comment there is more general and not directly related to the concern stated by the reviewer.

Page 12 and other points: I'm not sure "facilitated" is the best word when describing an increase in baseline artifact. Perhaps "increased the number of pipelines that showed a baseline artifact" or "increased the number of timepoints that showed a baseline artifact" or "increased decoding performance during the baseline period, suggesting a baseline artifact" would be better?

This point is well taken, and we highly appreciate the reviewer's alternative suggestions. We replaced the wording on page 12, line 310, and likewise on page 27, line 954.

References

¹ Bailey, N. W. *et al.* RELAX part 2: A fully automated EEG data cleaning algorithm that is applicable to Event-Related-Potentials. *Clin. Neurophysiol.* **149**, 202–222 (2023).